# Co-translational binding of importins to nascent proteins

Maximilian Seidel [1,2], Natalie Romanov[1], Agnieszka Obarska-Kosinska [1], Anja Becker[1], Nayara Trevisan Doimo de Azevedo[3], Jan Provaznik [3], Sankarshana R. Nagaraja [1], Jonathan J. M. Landry[3], Vladimir Benes [3] & Martin Beck [1,4] ✉

Various cellular quality control mechanisms support proteostasis. While, ribosome-associated chaperones prevent the misfolding of nascent chains during translation, importins were shown to prevent the aggregation of specific cargoes in a post-translational mechanism prior the import into the nucleoplasm. Here, we hypothesize that importins may already bind ribosome-associated cargo in a co-translational manner. We systematically measure the nascent chain association of all importins in *Saccharomyces cerevisiae* by selective ribosome profiling. We identify a subset of importins that bind to a wide range of nascent, often uncharacterized cargoes. This includes ribosomal proteins, chromatin remodelers and RNA binding proteins that are aggregation prone in the cytosol. We show that importins act consecutively with other ribosome-associated chaperones. Thus, the nuclear import system is directly intertwined with nascent chain folding and chaperoning.

Faithful protein biogenesis and the maintenance of a functional proteome poses a logistic burden for cells[1]. Errors in proteostasis result in protein aggregation, consequently leading to pathogenic phenotypes[2]. Therefore, it is crucial to ensure the quality of nascent proteins already in the vicinity of the ribosome. Nascent proteins are supported by an array of different co-translationally acting quality factors including nascent chain chaperones, nascent chain modifiers and translocation factors such as the signal recognition particle (SRP) and their protein complex partner subunits[3,4]. Ultimately, their synergistic action prevents intramolecular misfolding and ensures the reliable formation of stable multidomain arrangements[3,4].

Importins (also called karyopherins) are nuclear transport receptors (NTRs) that bind the nuclear localization sequences (NLSs) of their cargo in the cytoplasm and facilitate its passage through nuclear pore complexes (NPCs) into the nucleoplasm[5]. Moreover, importins contribute to proteostasis[6]. In vitro, importins inhibit the precipitation of basic, aggregation-prone cargoes by preventing their unspecific interaction with cytosolic polyanions such as RNA[7]. This has been shown for specific ribosomal proteins and histones[7]. The importin

transportin-1 (TNPO1 or Kapβ2) suppresses phase separation of RNA-binding proteins such as FUS and interference with importin-cargo binding causes cargo self-association and phase transitions[8–11]. Further, importins disaggregate NLS-bearing cargoes and even rescue neuro-degenerative phenotypes in vivo[8–11]. Similar chaperoning mechanisms may be relevant for TDP-43, TAF15, EWSR1, hnRNPA1, hnRNPA2, arginine-rich proteins and the spindle assembly factor TPX2[10,12,13]. These previous studies inferred post-translational chaperoning mechanisms for a limited number of individual cargoes. If importins bind to the nascent chains of their cargo in a co-translational manner to prevent aggregation, and if so, on which binding sites they generally act, remained unknown.

We reasoned that during the translation of many proteins that are destined to bind nucleic acids in the nucleus, basic patches are exposed as nascent chains. Protein folding in an RNA-rich environment such as the cytosol may thus critically depend on shielding of the respective patches. We therefore hypothesized that importins may bind to nascent chains. In this study, we systematically measured nascent chain association of all 11 importins in *Saccharomyces*

[1]Department of Molecular Sociology, Max Planck Institute of Biophysics, Frankfurt, Germany. [2]Faculty of Bioscience, Heidelberg University, Heidelberg, Germany. [3]Genomics Core Facility, European Molecular Biology Laboratory (EMBL), Heidelberg, Germany. [4]Institute of Biochemistry, Goethe University Frankfurt, Frankfurt, Germany. ✉e-mail: martin.beck@biophys.mpg.de

*cerevisiae*. We used selective ribosome profiling (SeRP)[14] to quantify the co-translational binding of importins to nascent proteins in a translatome-wide manner. Our approach led to the identification of a specific subset of importins that co-translationally associate with various cargoes, to the best of our knowledge many of them remained previously unidentified including different ribosome biogenesis factors, cell division machinery and regulators of transcription. We show that nascent chain binding by the chaperone Ssb1/2 frequently precedes cargo recognition by importins, in particular for nucleic acid binding proteins. We propose a model in which cargo complex formation is intertwined with nascent chain chaperoning to promote the faithful biogenesis of nuclear proteins. Our findings could have wider implications for our understanding of proteostasis in eukaryotes and of neurodegenerative disease.

## Results

### Selective ribosome profiling identifies the co-translational binding of importins to nascent cargoes

To systematically assess co-translational engagement of importins with nascent cargo (Fig. 1a), we used selective ribosome profiling (SeRP)[14,15]. This method relies on the affinity purification of co-translational interactors, in this case importins, whereby subsequent sequencing of ribosome-protected mRNA fragments serves as a quantitative proxy for positional chaperone association. It enables the quantification of co-translational binding of nascent chain chaperones to their substrates in a discovery mode for the entire translatome. Furthermore, SeRP systematically unravels the position of binding sites within the relevant open reading frames (ORFs) and thus provides their biophysical properties[16–19]. For affinity purification, we systematically tagged all 11 yeast importins (Supplementary Fig. 1a, b) with C-terminal twin-StrepII tags[20] using a scar-free cloning technique preserving the endogenous 3' untranslated regions[21]. We applied the primary amine-reactive cross-linker DSP to stabilize potentially transient interactions of importins which may be susceptible to RanGTP throughout lysis (Supplementary Fig. 1a, c) as previously described[15,16,22]. Previous systematic analysis has demonstrated that the stabilization of transient interactions by DSP increases reproducibility across replicates but does not affect chaperone binding patterns[16]. After RNase I digestion and enrichment of the ribosome-nascent chain complexes (RNC), we purified the respective co-sedimented importins from the RNCs (Supplementary Fig. 1d). We acquired SeRP data sets by sequencing the ribosome-protected fragments in four biological replicates for each of the 11 importins and a no-bait wildtype strain. We processed the sequencing reads as previously described to obtain the ribosome-protected footprints (Supplementary Fig. 1e)[14]. As expected, the footprints are much more prominent within the ORF in comparison to the respective 5' and 3' UTRs, showing that we have captured footprints from translating ribosomes on the respective mRNAs (Supplementary Fig. 1f). We note that the 3 nucleotide (nt) periodicity was blurred as compared to previous ribosome profiling experiments that did not use cross-linking[23,24]. This may be explained by a reduced accessibility of the mRNA for RNase I due to sterically hindrance by the cross-linker, which is consistent with the slightly increased ribosomal footprint length (Supplementary Fig. 1e), thus preventing an accurate registration of the A-site.

The resulting translatome-wide data set captures ribosome footprints for all mRNAs that are affinity-enriched for the respective importins. Pearson correlation between replicates was overall larger than between different conditions (Supplementary Fig. 2). To systematically identify potential hits, we developed a pipeline to short-list candidate profiles that consider both the IP and total translatome relative to a no-bait wildtype control (Supplementary Fig. 3). We used a manually curated list of cargoes of Srp1 and Kap95 from the literature as ground truth (Supplementary Fig. 3b and Supplementary Table 1) and an area under the curve (AUC)-value as a metric to identify co-translational cargoes. We note that our ground truth may contain

cargoes that only bind post-translationally, which would result in a conservative, over-estimation of the false discovery rate. Subsequently, we manually inspected all SeRP profiles short-listed by our analysis approach. Metagene plots before and after manual inspection (Supplementary Fig. 4) indicate that the 71 manually curated genes resemble high-confidence hits (Supplementary Fig. 5), in which the hits are statistically elevated over the background (Supplementary Fig. 6).

Our translatome-wide data allowed to systematically chart and to compare the co-translational cargo spectra of the different importins. Our approach was very complementary to previous studies that investigated post-translational cargo spectra[5,25–28], such that it very accurately identified heterodimeric interactions of importins instead of larger complexes but neglected those that only occur post-translationally. Pearson correlation of the AUC-values for all identified hits suggests a strong separation between the cargoes identified for the individual importins. The data obtained for the beta-type importins Kap114, Kap120, Kap121, Kap122, Mtr10, Sxm1, and Nmd5 largely correlated with negative no-bait controls (Fig. 1b). Indeed, very few or no cargoes were detected for this subset. We wondered if this might be related to the detection limit of our method; however, the abundance of importins and the number of identified hits did not correlate (Supplementary Fig. 7). For Srp1, Kap95, Kap123, and Kap121, 28, 27, 30 and 9 co-translationally bound cargoes were detected, respectively. The respective signal was distinct from negative no-bait controls (Fig. 1c). Strikingly, some of the co-translational cargoes of the Srp1-Kap95 correspond to literature-reported cargoes and show onsets at their literature-reported NLS (Supplementary Tables 1 and 2). Additional validation of Kap123 cargoes by using a previously reported RIP-qPCR approach[29–32] confirmed co-translational interactions of Kap123 with nascent chains (Supplementary Fig. 8).

In contrast to previous proteomic studies[33,34], we found little overlap between the set of cargoes identified for each importin (Supplementary Fig. 9), with the exception of Srp1 and Kap95 (see below). Among the identified hits, gene ontology (GO) analysis revealed enrichment for the nucleus and its sub-compartments underscoring that co-translational binding was specific for nuclear import cargoes (Fig. 1d). While cytoplasmic translation and translation termination were enriched in the Kap123-SeRP hit set, regulators of transcription by polymerase II, DNA repair and cell division were found in the Srp1-Kap95 set. We also found that both, Srp1-Kap95 and Kap123 shared enrichment for rRNA processing proteins (Fig. 1e). In contrast to Srp1-Kap95 and Kap123 which seemed to be distinct in their function, Kap121 was associated with processes within the nucleolus as well as the chromosome and telomeric regions. Taken together, this data pointed to a model in which Srp1-Kap95, Kap123, and Kap121 prominently associate with a specific set of cargoes in a co-translational manner, while other importins may preferably act post-translationally.

### Co-translational association of importins is enduring

Selective ribosome profiles allow for the visualization of importin binding events within an open reading frame, as shown for representative examples in Fig. 1f. In contrast to profiles previously obtained for the ribosome-associated Hsp70 chaperones Ssb1 and Ssb2 (Ssb1/2)[16,17], the chaperonin TRiC[16] or the signal recognition particle (SRP)[18,19], the importin-derived profiles suggested that once importin is bound to the nascent protein, it remained tethered (Fig. 1f). This is reminiscent of the co-translational interactions previously observed during protein complex formation[30,31]. This particular binding pattern may be due to the requirement of RanGTP for the dissociation of import complexes that is absent in the cytosol[35,36]. Thus, importins constitute an enduring chaperoning system that holds onto its substrates from synthesis in the cytosol until its release into the native context in the nucleus. A notable exception was Efr3 (Fig. 1f), which is annotated as a plasma membrane protein. Nevertheless, a strong signal was observed for the importin Sxm1 that binds to the nascent chain of Efr3 approximately at

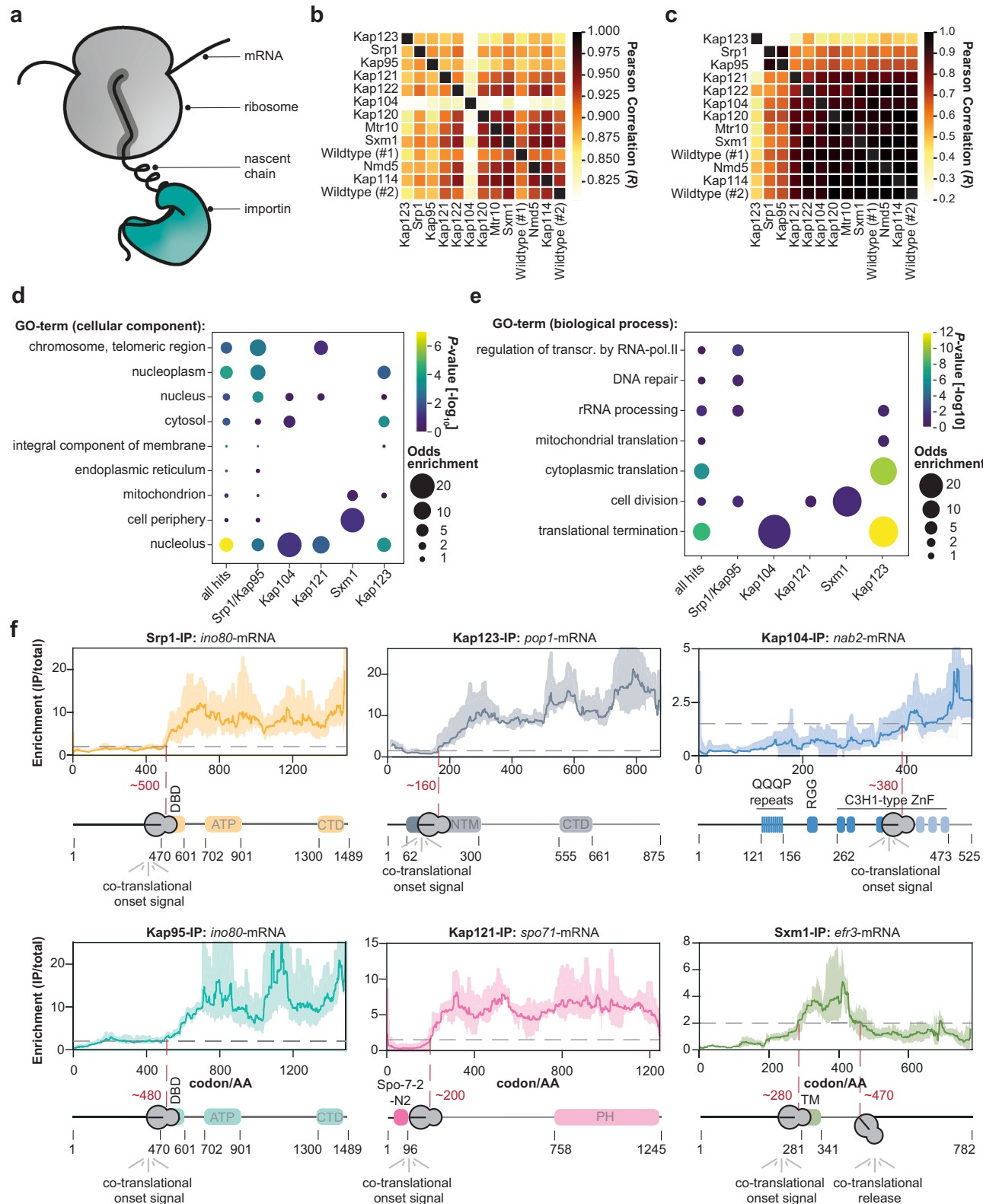

codon 280. Interestingly, it was released ~180 codons downstream, similar to the transient binding mechanisms of ubiquitous co-translational chaperones and the SRP.

## Srp1 and Kap95 mutually bind to nascent cargo

Srp1 (importin-alpha) and Kap95 (importin-beta) represent the classical nuclear import pathway in yeast[37]. In contrast to other beta-type importins that directly bind the NLSs of their cargoes, this classical

pathway requires importin-alpha as an adaptor. The interaction with importin-beta liberates the autoinhibitory NLS of importin-alpha from the NLS binding groove. This structural rearrangement results in the activation of the Srp1-Kap95 heterodimer that in turn binds to the cargo NLS[38]. To explore if Srp1 and Kap95 mutually bind to nascent chains, we compared the AUC-fold change of the Srp1- to the Kap95-SeRP experiments, which was highly correlated (Fig. 2a). 27 cargoes are common to both experiments (Fig. 2a, b). In addition, nascent Srp1

**Fig. 1 | SeRP of importins reveals co-translational binding to cargo. a** Scheme illustrating the co-translational binding of importins to a nascent chain. **b** Pearson correlation of the area under curve (AUC)-values of the selective ribosome enrichment profiles (IP/total) of 5855 genes quantified across the experiments. **c** Same as (**b**) but for the 71 manually curated cargoes. **d** Visualization of the Gene Ontology (GO)-enrichment for cellular compartments. While Srp1-Kap95 enriches chromosomal and telomeric regions, Kap123 shows enrichment for the nucleolus. Only significantly enriched GO-terms are shown (*P*-value < 0.1, not adjusted, (two-sided) Fisher Exact Test relative to all proteins quantified). **e** Same as (**d**) but for biological function. While Kap123 enriches for rRNA processing, cytoplasmic translation, and translation termination, Srp1-Kap95 rather enriches for cell division, DNA repair, and transcription regulation. **f** Representative SeRP profiles. In

most cases, importins associate with the nascent chain and subsequently remain bound, whereas Efr3 (Sxm1-SeRP) constitutes an exception. SeRP profiles (IP/total) are shown for the respective mRNA targets from *n* = 4 biologically independent replicates (solid lines are averaged across replicates; shades reflect largest to smalls replicate value interval). Gray dashed lines indicate an arbitrary threshold of 2 used for onset estimation (red dashed line). Note, that in the case of *nab2*-mRNA, a threshold of 1.5 was chosen. Domain annotation based on Pfam. transcr. transcription, RNA-pol. II RNA-polymerase II, IP immunoprecipitation, AA amino acid, DBD DNA binding domain, ATP ATP helicase domain, CTD C-terminal domain, NTM N-terminal motif, QQQP glutamine-rich region, RGG arginine-glycine-glycine domain, C3H1-type ZnF cysteine-cysteine-cysteine-histidine-type zinc finger domain, PH Pleckstrin homology domain, TM transmembrane domain.

itself was bound by Kap95 reflecting a co-translational protein complex formation (Fig. 2b, c). The respective SeRP profile showed an onset approximately at codon -150 (Fig. 2c), suggesting the interaction with Kap95 occurred once the synthesis of the importin-beta binding domain (IBB) was completed.

We therefore wondered if the onset observed for the Srp1 and Kap95 association occurred at similar positions within the relevant ORFs. SeRP profiles for both, Srp1 and Kap95 showed a pronounced N-terminal preference, contrasting other co-translationally acting importins (Fig. 2d and Supplementary Fig. 10). On the level of individual ORFs, simultaneous binding of Srp1 and Kap95 was observed (Fig. 2e and Supplementary Fig. 11), with very few exceptions, suggesting heterodimer formation prior to cargo binding. This finding suggested that the above-introduced mechanism of Srp1-Kap95 heterodimer activation, which has been elucidated by structural and biochemical analysis for a smaller set of substrates, appears to be broadly applicable to the co-translational formation of cargo complexes[27,38].

To address the functional relevance of the observed onsets, previous studies used genetic perturbation and biochemical assays[16,18,19,22,30,31]. We queried whether the respective peptides upstream of the onset would be sufficient for nuclear localization. Therefore, we generated GFP fusions of peptides within a 40 to 50 amino acid sequence window upstream of the onset, accounting for the emergence of the peptide from the ribosomal exit tunnel (Fig. 2f). While GFP without any fusion peptide was present throughout the entire cell, peptide fusions of 5 randomly selected cargoes (Srp1-Kap95: Ino80, Prp8; Kap123: Rps5, Pop1, Nup60) were sufficient for nuclear localization (Fig. 2g). In case of Pct1, for which a slightly shifted onset of Kap95 with respect to Srp1 was observed, we found that the N-terminally localized peptide showed a stronger nuclear enrichment (Fig. 2h).

### Prediction of classical NLSs in Srp1-Kap95 cargoes

To map putative cNLS in the proteins identified as hits, we ran AlphaFold-Multimer[39] structure prediction for pairs of Srp1 and consecutive overlapping fragments of the respective protein sequences (see Materials and Methods). For all hits, we obtained at least one prediction with a fragment occupying the NLS binding site of Srp1. Some hits contained two or more NLSs predicted with similar scores. The predicted NLSs frequently occurred at the N-terminal region in agreement with the N-terminal preference found within the SeRP data (Fig. 2d). All known NLS motifs were predicted with top scores (Supplementary Tables 2, 3) validating our procedure. The structural superposition (Fig. 3a) and structure-based sequence alignment revealed that most of the predicted motifs exhibit sequences resembling classical NLS (cNLS) motifs of Srp1 of either the monopartite (K-K/R-X-K/R) or bipartite (K/R-K/R-X$_{10-12}$-K/R$_{3/5}$) type (Fig. 3b)[40], whereby the linker region can be considerably longer. Some sequences, however, were very different from the sequence consensus or bound to the NLS binding site in the opposite direction and might correspond to false positive predictions or non-canonical NLSs. Altogether, these

results confirmed that the identified target proteins bind to Srp1 and allow for the prediction of the corresponding cNLS motifs.

### Kap123 cargoes act in early stages of ribosome biogenesis

Ribosomal proteins (r-proteins) are synthesized in the cytoplasm and are transported into the nucleolus where they associate with ribosomal RNAs (rRNA). Out of the 87 yeast r-proteins, 13 were detected in our screen, including 5 paralogous pairs. 12 out of 13 detected r-proteins co-translationally engaged with Kap123, the other one with Kap104 (Supplementary Fig. 5). We found that for the paralogous r-proteins, the selective ribosome profiles were identical in shape, and enrichments varied according to their paralog-specific expression levels[41] (Supplementary Fig. 12a). Moreover, none of the identified r-proteins overlapped with the substrate spectrum of known r-protein chaperones[42] (Fig. 4a), suggesting a unique functional role of Kap123 in ribosome biogenesis. Specifically, the identified r-proteins chaperoned by Kap123 are important in early stages of 60S and 90S ribosome biogenesis (Fig. 4b). The notion that Kap123 is relevant for early stages of ribosome biogenesis is further supported by other identified cargoes that include several early ribosome biogenesis factors such as e.g. Nug1, Noc2, Ecm16 (Fig. 1e, Supplementary Fig. 5)[43–46].

Interestingly, when we depicted the apparent onsets for the interaction of Kap123 into a mature 80S ribosome structure (PDB: 4V7R)[47] we noticed that they typically mapped to the C-terminally located structured domains of the respective r-proteins (Fig. 4c, d and Supplementary Fig. 10). We propose that this may have two functional benefits. First, Kap123 binding within structured domains may suppress potentially erroneous and toxic interactions of r-proteins outside of their native context. This would be in line with previous reports suggesting that the human homolog of Rps1a/b in complex with polyanions is resolubilized from by the human homolog of Kap123[7]. Second, the respective sites were engaged in various RNA contacts within the mature 80S ribosome suggesting an inaccessibility for importin binding once ribosome assembly is completed. A possible interpretation of this observation is that faithful structural rearrangements of r-proteins during biogenesis ultimately renders ribosomes invisible to the nuclear import system to prevent nuclear re-import.

### Ssb1/2 chaperoning occurs upstream of Kap123 binding

The majority of the identified r-proteins consist of a short, intrinsically disordered, and charged N-terminal patch followed by a globular and structured domain. Identification of potential binding peptides of Kap123 on these r-proteins (Fig. 4c) suggests Kap123-binding within the structured domains. We therefore wondered whether the broadly acting nascent chain chaperones Ssb1/2 that are known to suppress folding by binding the nascent chains[17] could act upstream of Kap123 to enable faithful Kap123 cargo recognition. We systematically analyzed the co-occupation of importin chaperoned nascent chains with Ssb1/2, and the Hsp60 TRiC/chaperonin using previously published data sets[16,17] (Supplementary Fig. 12b). We found that the nascent chains of all r-proteins bound by Kap123 and Kap104 are also captured by Ssb1/2, and in the case of Rpl8a, Rps1a/b, Rps5 and Rps9b also by

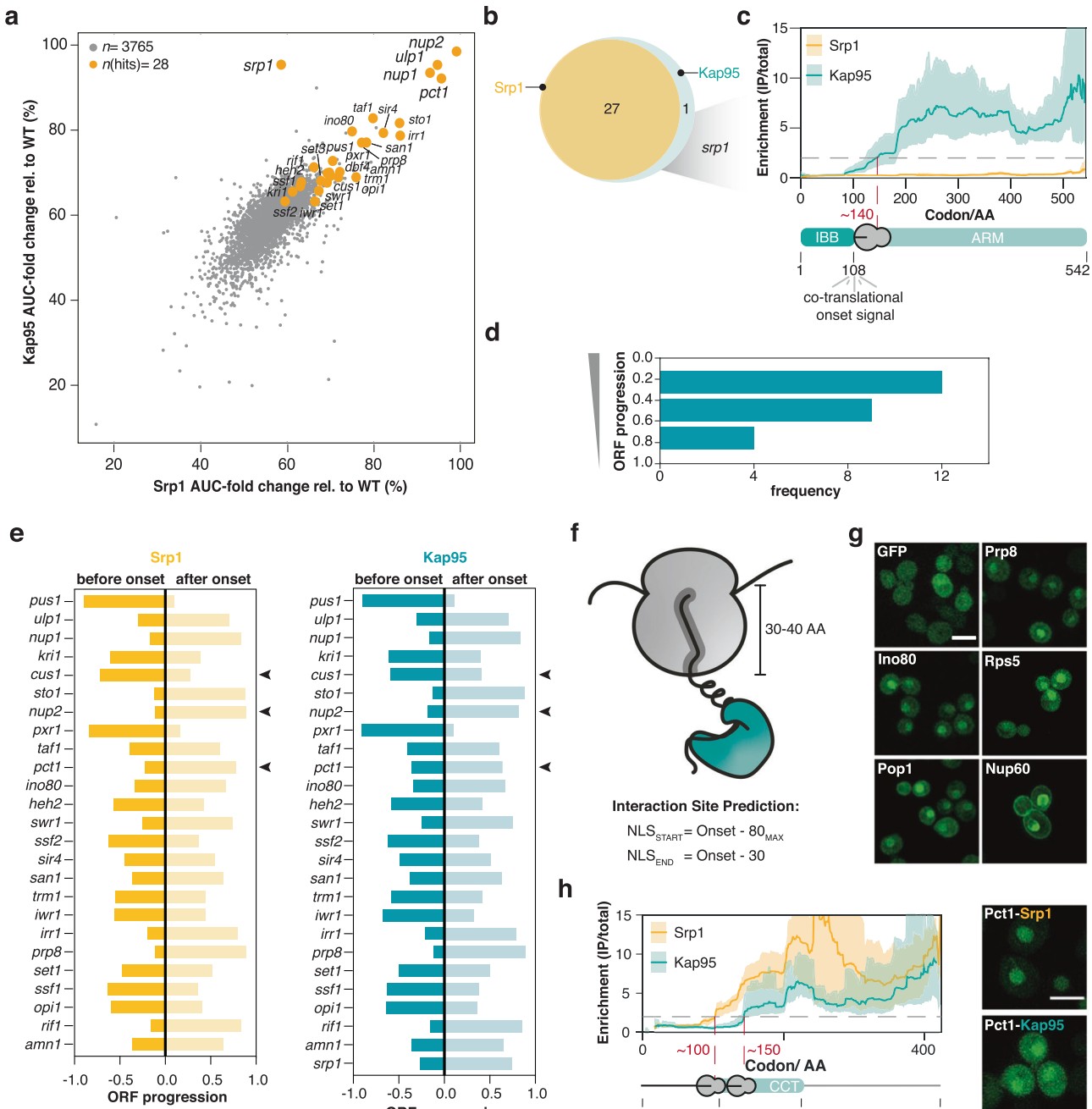

**Fig. 2 | Srp1 and Kap95 synchronously bind to nascent cargoes. a** Scatter plot of the AUC-values of genes quantified in Srp1 in comparison to Kap95 (detailed in Methods). The identified hits are highlighted in orange. **b** Venn diagram showing the overlap of the identified cargoes. **c**, Enrichment profile of *srp1*-mRNA in Srp1- and Kap95-SeRP experiments. Kap95 binds to nascent Srp1 at codon ~140 corresponding to the release of the entire IBB domain. SeRP profiles (IP/total) are shown for the respective mRNA targets from $n = 4$ biologically independent replicates (solid lines are averaged across replicates; shades reflect largest to small replicate value interval). Gray dashed lines indicate an arbitrary threshold of 2 used for onset estimation (red dashed line). **d** Distribution of onsets shows an N-terminal preference. **e** Comparison of the onsets observed in the Srp1- and Kap95-experiments.

Arrowheads indicate 3 slightly divergent cases. **f**, **g** Peptides upstream of the observed onsets are sufficient for nuclear localization of the respective GFP fusion proteins. **g** Representative confocal images for GFP fusions with peptides from the indicated cargoes. Scale bar: 5 μm. **h** SeRP profiles as in (**c**) but for nascent Pct1 indicate a slightly shifted onset of Srp1 and Kap95; the N-terminally localized peptide shows stronger nuclear localization apparent in confocal slices (as in **f**). For (**g** and **h**), imaging was conducted twice ($n = 2$) for at least 100 cells. Source data are provided as a Source Data file. IP immunoprecipitation, AA amino acid, IBB importin-beta binding domain, ARM armadillo repeat, NLS nuclear localization sequence, GFP green fluorescent protein, CCT choline-phosphate cytidylyltransferase.

TRiC. Across the Ssb1 co-chaperoned r-protein SeRP profiles, we found that Ssb1 binding temporally preceded Kap123 binding (Fig. 5a, b, and Supplementary Fig. 12c). A notable exception was Rpl28 that may not require Ssb1/2-binding (Figs. 4d, 5c). Previous mass spectrometry data indicated that a subset of the yeast proteome is rendered aggregation-prone in the absence of Ssb1/2[48]. While Ssb1/2 substrates were depleted in nascent Srp1-Kap95 or Kap121 cargoes as compared to the nuclear proteome, they were enriched in the corresponding nascent Kap123 cargoes (Supplementary Fig. 12d). Interestingly, nascent Kap123 cargoes very frequently aggregated in the absence of Ssb1/2 (Supplementary Fig. 12d), suggesting that both processes are intertwined. Taken together, this analysis suggested a Ssb1/2-importin

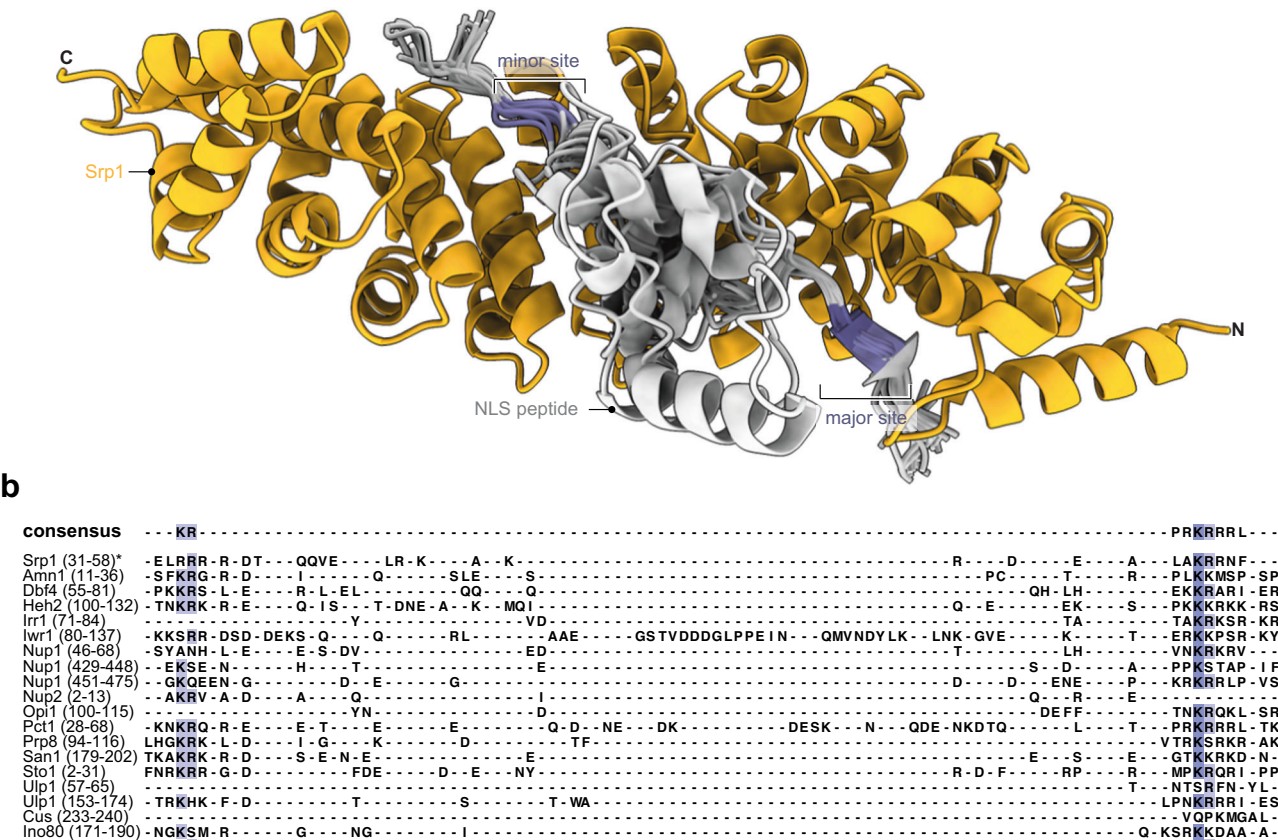

**Fig. 3 | Co-translationally bound Srp1 cargoes have predicted cNLSs. a** Srp1 (AA 70–512) structure (orange) with peptides modeled using AlphaFold-Multimer. Only predictions i) with the ipTM+pTM score > 0.7, ii) bound to the canonical NLS-binding site in Srp1, iii) bound in the N- to C-terminus orientation as known from crystal structures of Srp1-NLS are shown (PDB: 1WA5)[82]. The entire data set is listed (Supplementary Data 1). **b** Structure-based sequence alignments derived from subpanel (**a**), display hallmarks of cNLSs. Regions of the target sequences are shown in the alignment indicated in parentheses (in AA). Peptides are colored by sequence conservation. *inhibitory peptide of Srp1 (PDB: 1WA5)[82].

handover mechanism for substrate recognition (Supplementary Fig. 12e). Since Ssb1/2 is thought to retain hydrophobic and positively charged nascent peptides in a linear, degenerated form[16,17], it may facilitate co-translational importin cargo recognition.

## Co-translational importin binding sites have distinct biophysical properties

While small molecules can diffuse across NPCs, larger molecules above ~30–50 kDa require active transport[49]. We therefore asked if co-translationally associated cargoes were above this size threshold. Our analysis showed that the majority of the Srp1-Kap95 chaperoned cargoes exceed the size threshold and thus depend on active import. This was less pronounced for the Kap121 cargoes and stood in strong contrast to the Kap123 cargoes. In the latter set, small proteins with a median size of only ~30 kDa were particularly enriched suggesting that the majority of which could also passively enter the nucleus if they were not bound to Kap123 (Supplementary Fig. 13a). Concomitantly, many of the Kap123 cargoes were highly abundant and synthesized with an exceptionally high translation rate (Figs. S13b, c). These findings point to a model in which co-translational cargo complex formation may mask potentially harmful biophysical properties, in particular of Kap123 cargoes.

Gene Ontology analysis indicated that many co-translational cargoes encode for ribonucleic acid binders (Fig. 6a). The apparent onsets of importins within the ORFs of co-translational cargoes were enriched for specific types of domains, namely tRNA methyltransferases as well as DNA-, RNA- or histone processing domains. To a much lesser extent, they occurred at sites of protein-protein interactions or membrane association (Fig. 6b). Since nucleic acid binding domains are often charged, we assessed the isoelectric point (pI) of the proteins detected by our screen. In comparison to the entire nuclear proteome, co-translationally bound cargoes had higher pI-values and were enriched for lysine and arginine, in particular Kap123 cargoes (Fig. 6c, Supplementary Fig. 13d). Interestingly, co-translational cargoes of importins bear strong positive charges as compared to the substrates of the ubiquitous chaperones Ssb1/2 and TRiC (Supplementary Fig. 13e), stressing the unique role of importins. The local amino acid signature upstream of the observed onsets for Srp1 displays a significant enrichment for the positively charged residues lysine and arginine. This enrichment appears less accentuated for Kap123 (Fig. 6d). However, this may be due to the generally high lysine and arginine content within its cargoes (Supplementary Fig. 13d). At last, we analyzed if the elongation fidelity is affected by the compositional bias at importin binding sites. We noticed an increased ribosomal occupancy upstream

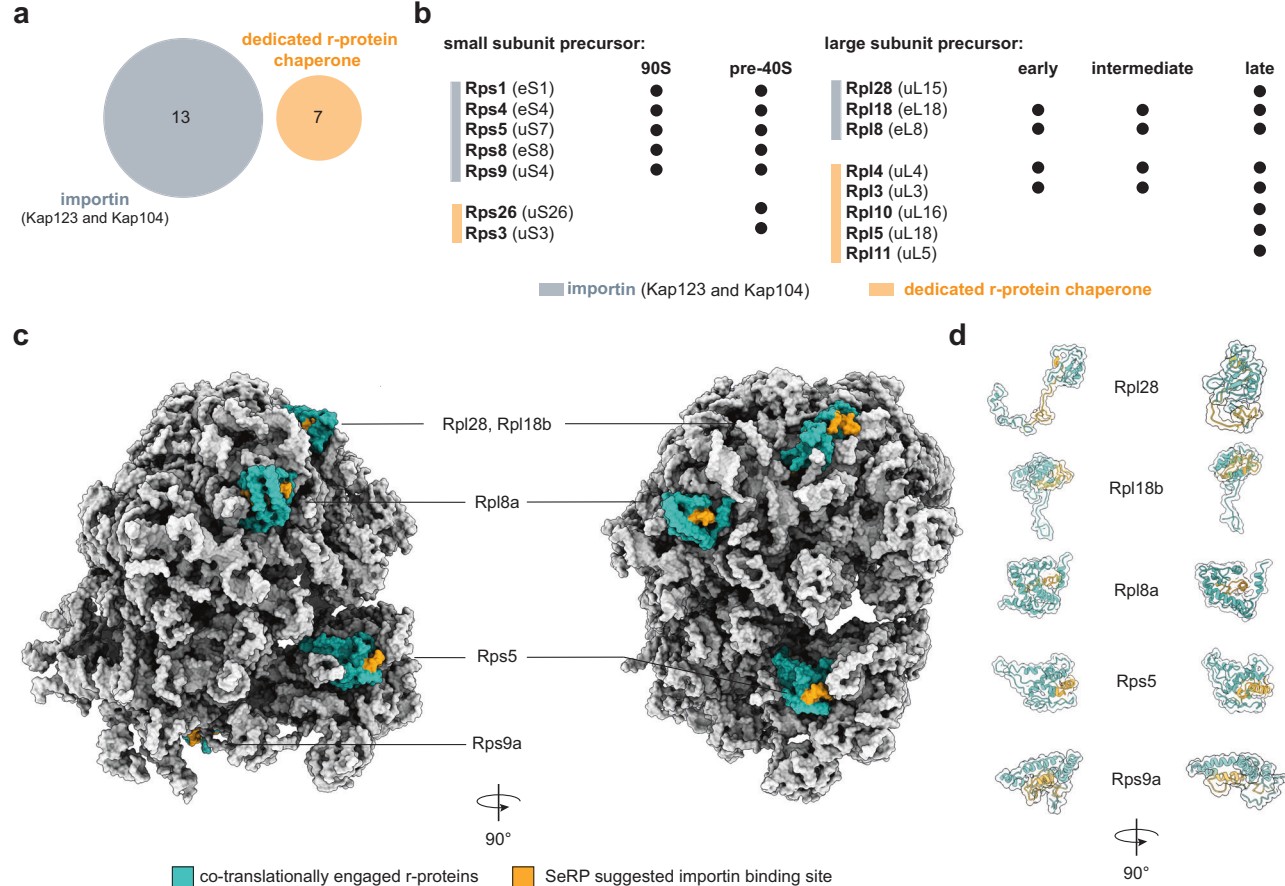

**Fig. 4 | Engagement of Kap123 with nascent chains reveals novel aspects in ribosome biogenesis. a** Venn diagram of cargo overlap between importins and dedicated r-protein chaperones (Tsr2, Yar1, Acl4, Rrb1, Sqt1, and Syo1)[42]. The importins Kap123 and Kap104 chaperone unique r-proteins which else would not be covered by the substrate spectrum of r-protein chaperones. **b** Importin chaperoned r-proteins are required in early ribosome biogenesis. **c** Potential Kap123 and Kap104 binding moieties are buried within mature 80S ribosome. Co-translationally engaged r-proteins are highlighted in turquoise with their potential importin binding site highlighted in orange. Peptides highlighted in orange represent −50 to −30 of the onsets. **d** same as for (**c**), but represented as cartoons. Depicted structures can be accessed at the PDB: 4V7R[47].

to the observed onset suggesting a decrease in elongation prior to importin binding (Fig. 6e, f). Interestingly, the charged lysine and arginine residues that are enriched at the observed onsets of many cargoes, are not only a hallmark of the NLS motif but they are also associated with less abundant tRNAs[50] and thus likely reduce translation speed to warrant importin binding. These findings generalize the basic chaperoning function of importins for numerous cargoes and provides more detailed insights about the chaperoned cargo sites and their biophysical properties.

## Discussion

Taken together, our data point to the following model (Fig. 6g): Importins bind to the nascent chain of many cargoes during protein synthesis. This mechanism is particularly prominent for basic nuclear and r-proteins that are primarily substrates for Srp1-Kap95, Kap121, and Kap123. Prior to co-translational engagement of importins, many lysins and arginines, which are often constituents of nucleic acid binding domains, are synthesized causing an intrinsic reduction of the translational speed due to their rare codon usage. This decrease in translational fidelity may be beneficial for faithful importin association. In some cases, we even observe a direct handover of the nascent chain between temporarily bound Ssb1/2 and importins. This handover might be necessary for cargo recognition by importin in particular for proteins whose importin binding sites become inaccessible in the ternary structure and that could be kept unfolded by Ssb1/2 to promote faithful co-translational importin-cargo interaction.

Predicted and experimentally characterized NLSs indicate that in some cases, the observed onset may be shifted downstream to the physical importin binding site. This may be explained by the modulation of the availability of the binding site by other nascent chain binders as exemplified for Hsp70. Alternatively, domain recognition of importins may explain this phenomenon as exemplified by a recent structural study[51]. Furthermore, the structure of some importins, in particular Kap123, are shaped such that they may warp around their cargoes to protect them from their environment[51,52]. This binding mode could be reminiscent to the activity of trigger factor that co-translationally cages substrates to regulate aggregation-prone regions[53].

We propose that the co-translational nuclear import complex formation shields positively charged patches early during biogenesis, in an RNA-rich environment. This may be particularly relevant for nucleic acid binding domains that otherwise may be aggregation-prone in the cytosol. These insights strengthen the notion that importins are part of a basic client chaperone network. It was previously shown that importin 4 (yeast: Kap123) chaperones the basic RPS3A (yeast: Rps1a)[7] that otherwise aggregates in the presence of tRNA. Our study highlights that this protection of Rps1a is already established co-translationally. This chaperoning presumably lasts until the nuclear entry of the import complex. In the nucleoplasm, it becomes exposed to RanGTP and the cargo is released into the destined biophysical environment to engage with native interaction partners. Some of the cargo complexes may rely on alternative release

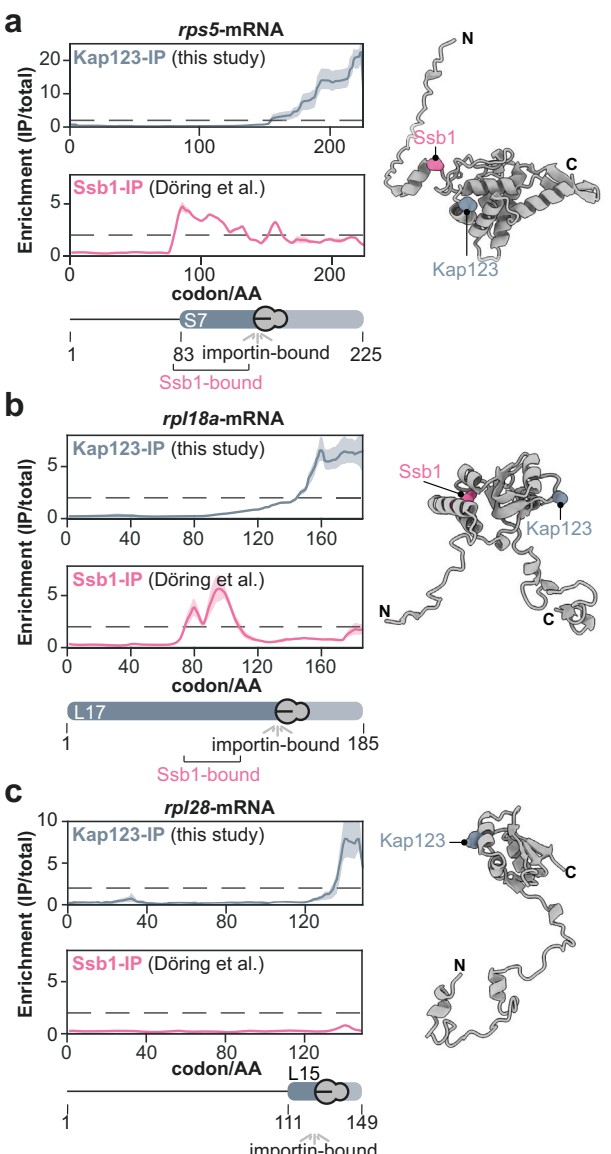

**Fig. 5 | Nascent chaperoning and cargo complex formation of r-proteins.** SeRP profiles and AlphaFold models of (**a**) Rps5, (**b**) Rpl18a, and (**c**) Rpl28. **a, b** The SeRP profiles reveal, that for nascent Rps5 and Rpl18a, Ssb1 first binds to the nascent chain. Subsequently, it is released prior to Kap123 binding. **c** In contrast, Rpl28 does not require Ssb1 chaperoning. Nevertheless, Kap123 binds C-terminally. The folds exemplify the structural organization of many r-proteins. A nascent r-protein recognized by Kap123 contains an N-terminally disordered, charged patch and a consecutive structured domain. SeRP profiles (IP/total) are shown for the respective mRNA targets from $n = 4$ biologically independent replicates (solid lines are averaged across replicates; shades reflect largest to smalls replicate value interval). Gray dashed lines indicate an arbitrary threshold of 2 used for onset estimation. SeRP for Ssb1[17] represents profiles generated from $n = 2$ biologically independent replicates. AlphaFold models of r-proteins were obtained from the AlphaFold database (https://alphafold.ebi.ac.uk/)[83,84]. C-terminal binding sites (40 AA downstream of onset) are labeled for Ssb1 and Kap123 within the structures, respectively. IP immunoprecipitation, AA amino acid.

cues, such as the histone dimer H2A•H2B, the SUMO-deconjugating enzyme Ulp1, the mRNA binding protein Nab2, Nab4, and Npl3, or the ribosomal protein eS26[52,54–57].

Beyond the co-translational proteostatic function of importins, our study extends the known spectrum of cargoes chaperoned by importins beyond ribosomal proteins, histones, and some RNA binding proteins[7,10]. We found 71 unique co-translational substrates

(summarized in Supplementary Fig. 5), many of them accounting for previously undescribed nuclear transport cargoes. Surprisingly, histones and many of the r-proteins did not enrich co-translationally. This may be due to the action of additional and very specialized chaperone networks that may protect them from misfolding in a co-translational fashion[42,58,59]. Although importins have been shown to be partially redundant in function and their cargo spectrum[33,37], our data indicate a rather low redundancy in the co-translational binding capacity (Fig. 1b, c and Supplementary Fig. 9). It has been previously hypothesized that local translation of nucleoporins at nuclear pores could be mediated by importins[60], but a direct interaction of importins with nascent nuclear proteins had not been shown. We find that the Srp1-Kap95 heterodimer, Kap123, Kap121 and to lesser extent Kap104, Sxm1, Mtr10 and Kap122 co-translational act on nascent chains, while Kap114, Kap120, and Nmd5 did not show any significant binding under the conditions tested (Fig. 1b, c). It remains yet unclear why only some but not all importins act co-translationally. Although the sequence conservation is considerably low across the 11 yeast importins, they are unified by their low isoelectric point (pI = 4.0–5.0), helical HEAT-repeat rich solenoid or superhelical structures and a negatively charged NLS binding pockets[5]. Interestingly, the importins detected as co-translational binders by our study frequently bind to nuclear proteins that are important for maintaining cellular viability under optimal growth conditions. In contrast, cargoes that were previously described for the other subset of importins, but remained undetected in our experiments, frequently form import complexes with co-chaperones or transcription factors that are activated or translocated into the nucleus upon stress conditions, e.g. Kap114 that is indispensable under saline stress[61]. It will thus be interesting to investigate such interactions under permissive conditions in the future.

Overall, our findings suggest a role of importins as proteostatic safeguards for nascent nuclear proteins but also open up novel perspectives on previous findings that associated importins with biomolecular condensation. For example, Kap123 and some of the here identified co-translational cargoes (e.g. Nug1 and Noc2) were reported to phase separate upon heat shock[62]. We speculate that recruitment of Kap123 may ensure reversibility by protecting the RNA-binding patches of cargoes in the respective RNA containing granules. Furthermore, importin-alpha, importin-beta, and the Kap121 homolog importin-β3 were found to co-translationally associate with Nup358-granules that manufacture NPCs in early fly development[63]. Most importantly, importins were attributed to counteract neurodegenerative disease by enhancing the solubility of nuclear proteins associated with pathological features such as FUS and TDP-43[6]. Some of the genes identified in *S. cerevisiae* in our study are known to drive neurodegeneration in humans. Among these genes are *prp8* that is associated with retina pigmentosa[64], *efr3* that is mutated in autism spectrum disorder[65], and *taf1* that if mutated can cause intellectual disability[66]. Our study demonstrates that the solubility of such proteins may be enhanced co-translationally to prevent the exposure of aggregation-prone ribonucleic acid binders prior to their full accessibility to the cytoplasm.

## Methods
### Yeast strain design
Scar-free C-terminally twin-StrepII-tagged importin strains were obtained by homologous recombination using the MX4 blaster cassette[21]. For this method, the MX4 blaster cassette was amplified with gene-specific overhangs to recombine immediately after the endogenous STOP codon of the gene of interest. These PCR products were then transformed into BY4741 and selected on YPD-high phosphate plates containing 300 μg/mL hygromycin B (ForMedium) and 3 g/L potassium dihydrophosphate (monobasic). In the second round of transformations, MX4 blaster cassette was substituted with a codon-optimized StrepII-tag with flanking gene-specific overhangs as

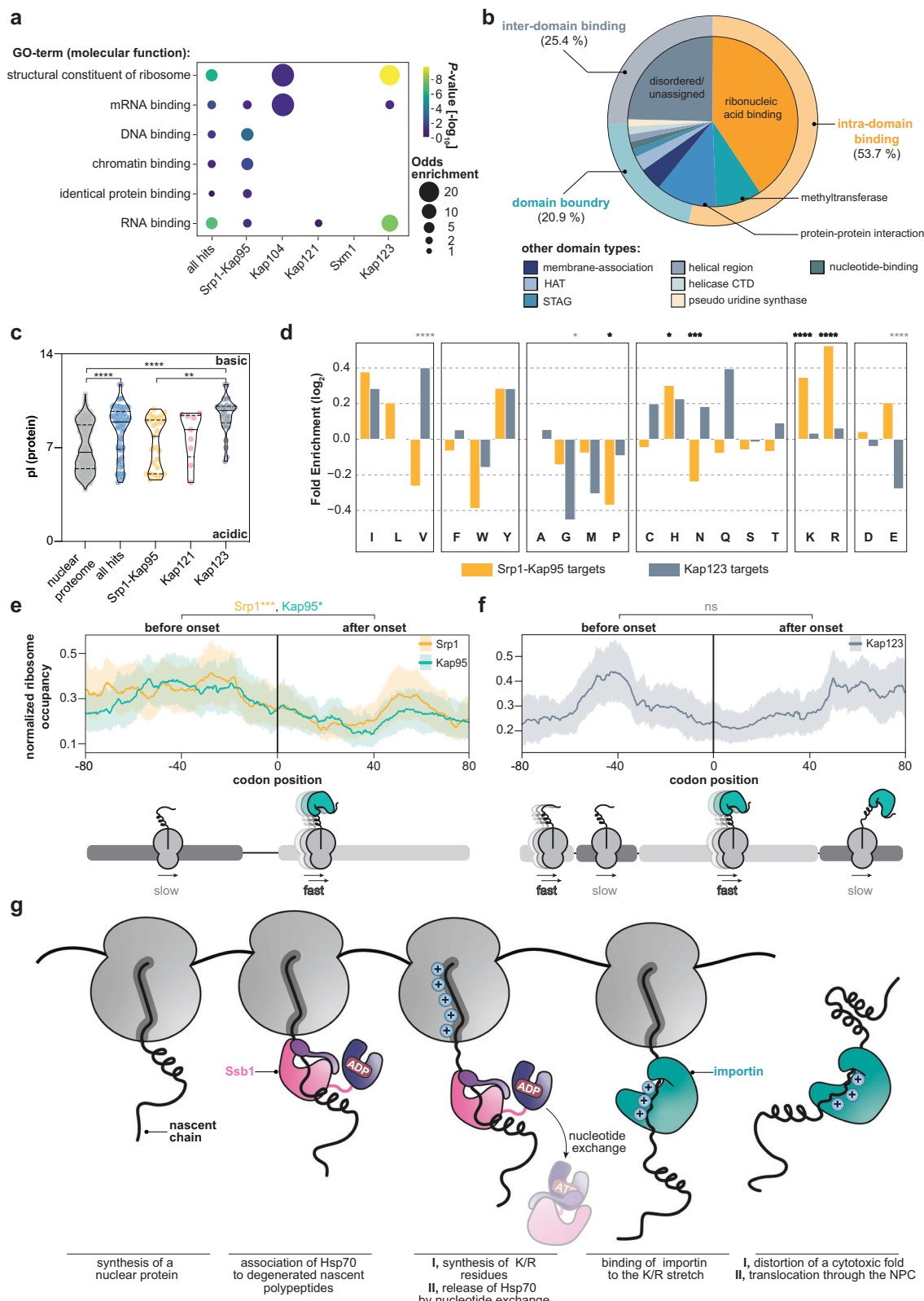

previously described[30]. For this transformation, the cells were grown in low phosphate YPD[21] and then transformed. Clones were selected on YP-galactose plates. We validated twin-StrepII-tag insertion by PCR.

For the NLS-GFP constructs, we obtained plasmids by Gibson Assembly (NEB) of the NLS flanked with 20 bp overhangs and a pRS316 containing a *tef1*-promoter:: GFP:: *cyc1*-terminator. 5 ng of plasmid was transformed into BY4741 and the cells were selected on synthetic

complete uracil drop out plates (ForMedium). All yeast strains generated in this study are listed in Supplementary Table 3.

## Selective ribosome profiling

Selective ribosome profiling was conducted as previously described[4,14,15,30]. Briefly, 800 mL of *S. cerevisiae* containing one of the eleven Twin-StrepII-tagged importins were inoculated with a starting

**Fig. 6 | Importins protect positively charged ribonucleic acid binding domains.** **a** Visualization of the Gene Ontology (GO)-enrichment for molecular functions. Co-translationally targeted cargo shows strong enrichment for DNA-, RNA-, chromatin-, protein-binding properties, and structural proteins of the ribosome. Only significantly enriched GO-terms are shown (*P*-value < 0.1, not adjusted, Fisher Exact Test (two-sided) relative to all proteins quantified). **b** Onsets are frequently observed at ribonucleic acid binding sites. Apparent onsets were mapped as described in Methods and classified according to their annotated function. **c** Importins capture preferentially positively charged nascent cargoes. While the pI-values of nuclear proteins are distributed bimodally, nascent cargoes are shifted towards high pI-values. Violin plot shows median and quartiles. Figure shows $n = 859$ (nuclear proteome), $n = 71$ (all hits), $n = 27$ (Srp1-Kap95), $n = 9$ (Kap121) and $n = 30$ (Kap123) individual proteins. ****$P = 3.53 \times 10^{-6}$ (nuclear proteome, all hits); ****$P = 5.72 \times 10^{-9}$ (nuclear proteome, Kap123); **$P = 0.0080$ (Srp1/Kap95, Kap123). Non-parametric Mann–Whitney *U* test (two-sided). Source data are provided as a Source Data file. **d** Amino acid enrichment in apparent onsets for nascent Srp1-Kap95 and Kap123 cargoes as compared to the full-length proteins (Fisher Exact Test; two-sided). *$P = 0.0101$ (Srp1-Kap95; proline); *$P = 0.0152$ (Srp1-Kap95; histidine); ***$P = 0.0015$ (Srp1-Kap95; asparagine); ****$P = 6.16 \times 10^{-9}$ (Srp1-Kap95; lysine); ****$P = 1.32 \times 10^{-4}$ (Srp1-Kap95; arginine); ****$P = 3.93 \times 10^{-4}$ (Kap123; valine). *$P = 0.0489$ (Kap123; glycine). ****$P = 2.42 \times 10^{-4}$ (Kap123; glutamate). **e, f** Metagene analysis of the ribosome occupancy at apparent onset sites for nascent Srp1-Kap95 (**e**) and Kap123 (**f**) cargoes. Prior to the onset, an increased occupancy is observed, implying a decrease in elongation. ***$P = 0.0028$ (Srp1: before, after onset); *$P = 0.0113$ (Kap95: before, after onset); ns = 0.8642 (Kap123: before and after onset); two-sided *t*-test across entire distribution. Solid lines represent averaged profiles; shades reflect a 95% confidence interval. For (**c–f**), ns $P > 0.05$, *$P < 0.05$, **$P < 0.01$, ***$P < 0.005$, ****$P < 0.001$. **g** Conceptual model of nascent cargo complex formation. As the nascent chain emerges from the ribosome, it may be bound by ubiquitous chaperones (e.g. Ssb1) that temporarily chaperone structured patches. Once released, importins nascently form cargo complexes and shield-charged patches. GO gene ontology, HAT histone acetyltransferase, CTD C-terminal domain, pI isoelectric point.

OD(600) of 0.035 in YPD. Cells were grown at 30 °C and 160 rpm to an OD(600) of 0.5–0.6. Afterward, cells were harvested by rapid filtration onto 0.45 μm nitrocellulose (Biorad), scraped off the membrane, and flash frozen in liquid nitrogen. Subsequently, cells were supplemented with 2.4 mL of lysis buffer (20 mM Hepes-KOH, pH 7.5, 140 mM KCl, 10 mM MgCl₂, 1 mM PMSF, 0.01% IGEPAL, 0.1 mg/mL CHX, 1 tablet of cOMPLETE protease inhibitor per 50 mL) and lysed in a CryoMill (Retsch) at 30 Hz for 2 min.

Next, 2.4 mL of lysis buffer was placed into a 5 mL beaker on a magnetic stirrer and supplemented with 30 μL of 250 mM DSP. Gradually, the first half of the lysate powder was stirred in until thawed. Supplementation of DSP was repeated and the second half of lysate was added. Crosslinking was carried out for 10 min at room temperature while constantly stirring. The reaction was quenched by adding 400 μL of 2 M Tris-HCl pH 8.0. Next, the crosslinked lysate was cleared at 15,000 *g* for 3 min at 4 °C. The supernatant was transferred and absorbance at 260 nm of a 1:100 dilution was measured. To generate ribosome-protected footprints, 60 U [RNase I]/ absorbance [RNA] unit was added to each sample. RNase I digest was carried out by end-to-end mixing at 4 °C for 30 min. RNase I reaction was quenched by adding 200 U of Superase•In. The supernatant was then applied onto a 25% sucrose cushion (25% w/v sucrose, 20 mM Hepes-KOH, pH 7.5, 140 mM KCl, 10 mM MgCl₂, 0.01% IGEPAL, 0.1 mg/mL CHX, 1 tablet of cOMPLETE protease inhibitor per 50 mL). Ribosomes were pelleted at 150,000 *g* for 2.5 hr. Then, the supernatant was discarded and the pellet was resuspended in 1 mL wash A buffer (20 mM Hepes-KOH, pH 7.5, 140 mM KCl, 10 mM MgCl₂, 0.01% IGEPAL, 0.1 mg/mL CHX, 1 tablet of cOMPLETE protease inhibitor per 50 mL). 100 μg of RNA was collected after resuspension representing the "total translatome". This RNA was supplemented with 750 μL of 10 mM Tris HCl pH 8.0, and frozen in liquid nitrogen until further purification (see below).

The residual resuspended pellet was added to 250 μL of pre-equilibrated Streptactin resin (IBA). Additionally, 30 μL of BioLock (IBA) was added. Affinity purification of the Twin-StrepII tagged importin was carried out by end-to-end mixing for 1 hr at 4 °C. Next, beads were centrifuged at 500 *g* for 5 min and the supernatant was removed. Subsequently, beads were washed three times in 1 mL of wash A, each time applying a 1 min end-to-end mixing step. Last, beads were washed in wash B buffer (20 mM Hepes-KOH, pH 7.5, 140 mM KCl, 10 mM MgCl₂, 10% v/v glycerol, 0.05% IGEPAL, 0.1 mg/mL CHX, 1 tablet of cOMPLETE protease inhibitor per 50 mL) first time for 1 min and the second time for 4 min using end-to-end mixing.

After washing, the supernatant was removed and the beads were resuspended in 500 μL 10 mM Tris HCl pH 8.0 and 40 μL of 20% SDS and gently mixed. Next, 750 μL of pre-warmed phenol-chloroform-isoamyl alcohol (PCI, 65 °C, Invitrogen) was added. This reaction was incubated at 65 °C, 1,400 rpm for 5 min and was immediately snap cooled on ice for 10 min prior to centrifugation at 15,000 *g* for 10 min. Once centrifuged, the aqueous phase was transferred into a new tube and supplemented with 750 μL of PCI. The mixture was occasionally vortexed at room temperature for 5 min and centrifuged again. To remove residual PCI, a diethyl ether (Sigma Aldrich) wash was carried out. The residual organic solvent was evaporated using a Speedvac (Eppendorf).

RNA was precipitated by adding 3 M NaOAc, pH 5.5 to obtain a final concentration of 0.3 M, 2.5 μL of Glycoblue (Invitrogen), and equal volumes of isopropanol. Precipitations were vigorously vortexed and incubated at −80 °C overnight. RNA was pelleted at 15,000 *g* and 4 °C for 90 min. the supernatant was removed and the pellet was washed three times with 70% ethanol. The resulting RNA pellet was dried in the Speedvac (Eppendorf) and resuspended in 20 μL of 10 mM Tris HCl, pH 8.0.

## Library preparation
Purified RNA and the 5′ 6FAM labeled RNA marker were mixed with an equal volume of 2 × RNA loading dye (Thermo Scientific) and heat-denatured at 80 °C for 2 min and immediately put back on ice before loading onto a pre-warmed 15% denaturing urea polyacrylamide gel electrophoresis (PAGE) (Carl Roth). Gels were run for 3.5–4 hrs at 16 W until bromophenol blue emerged, disassembled, and stained in Sybr-Gold (Invitrogen) according to the manufacturer's protocol and imaged using the Amersham Typhoon (GE Healthcare). RNAs at a size of 26–34 nt were isolated and crushed and soaked in 500 μL of Tris-HCl pH 8.0 at 70 °C for 10 min and constant and vigorous shaking at 1400 rpm. The slurry of gel pieces and buffer were then transferred into a Spin-X cellulose acetate column (0.22 μm; Corning), and centrifuged for 15 min and 4 °C at 15,000 *g*. Flow through containing the RNA of interest was supplemented with 50 μL 3 M NaOAc, 2.5 μL Gly-coblue co-precipitation agent (Invitrogen), and 500 μL isopropanol. The precipitation reaction was vortexed and incubated overnight at −80 °C. RNA was precipitated by centrifugation at 15,000 *g* for 90 min, washed three times with 70% ethanol, and was finally resuspended in 15 μL nuclease-free water.

Following their purification, the ribosome footprints were dephosphorylated on their 5′ and 3′ end using 1 × FastAP buffer, 2 U FastAP (Thermo Scientific), and 20 U RiboLock (Invitrogen). Dephosphorylation was carried out for 15 min at 37 °C at 600 rpm. The reaction was quenched by heat-inactivation of the FastAP at 75 °C for 5 min. Consecutively, 5′ ends were phosphorylated by incubating the reaction with 20 U polynucleotide kinase (PNK; NEB), 1 mM ATP (Thermo Scientific), 1 × PNK buffer (NEB), and 20 U RiboLock to prevent RNA degradation. The phosphorylation was carried out at 37 °C for 30 min.

Once the RNA was appropriately modified, integrity and concentration of the ribosome footprints were determined using the RNA Pico 6000 Assay Kit of the Bioanalyzer 21000 system (Agilent Technologies). 1 ng of RNA was used as input for library preparation using the NEXTflex Small RNA-seq Kit v3 (Perkin Elmer). Following this procedure, the quality of the libraries was assessed using the DNA High Sensitivity kit (Agilent Technologies), and the concentration of the library was determined using the Qubit DNA High Sensitivity kit using the Qubit 2.0 Fluorometer (Life Technologies). The concentration of each library was calculated and pooled at an equimolar amount. These multiplexed pools were finally purified with SPRI select beads with an excess of 1.3 × of beads (Beckman Coulter). The purified ribosome profiling library pool was loaded onto the Illumina sequencer Next-Seq2000 and sequenced uni-directionally, yielding ~1379 million reads with a size of 72 bases.

### Sequence processing

Data from the NextSeq2000 was processed following instructions published in Galmozzi et al.[14] and the script suite provided within the aforementioned instructions (https://doi.org/10.5281/zenodo.2602493). Reads were cleaned and trimmed using cutadapt (v2.3)[67]. Non-coding RNAs of *Saccharomyces cerevisiae* were filtered and excluded for further analysis using a non-coding reference genome (R64-1-1.ncrna). Reads encoding for coding RNAs were mapped to a *Saccharomyces cerevisiae* (R64-1-1) reference genome using tophat2 (v2.0.10). Ribosome centering was carried out as described within the script suite (https://doi.org/10.5281/zenodo.2602493).

Output files of script A which contain the numbers of reads per genomic position were extracted and used as input for our MATLAB scripts (v9.7.0.1296695 (R2019b) Update 4) as described in Seidel et al.[30] and for our novel python pipelines.

### Ribosome profiling data analysis and target identification

Genes were mapped to the S288C (R64) reference as downloaded from SGD[68] (*S288C_reference_sequence_R64-1-1_20110203.fsa*); introns and exons were mapped to sequence data using *sacCer3.ensgene.gtf* as extracted from Ensembl[69]. The following analysis is based on normalized read counts as the output of the aforementioned sequence processing. Per experiment (IP, and total, alike) replicates were averaged for each nucleotide. In each experiment, a mean correlation value (Pearson) was calculated for each gene across replicates; this is later used for quality filtering of potential pulldown targets (threshold 0.6). Gene ribosome profiles for each replicate, and averaged, were smoothed using a sliding window of 100 nucleotides (calculation of moving average) using the *pandas* (Python) package[70]. The orientation of the respective strand was considered; introns were not removed for smoothing. From these smoothed profiles in pulldown experiments (IP) and respective controls (total), the area under the curve (AUC) was calculated (excluding introns) using the trapezoidal rule as implemented in the *numpy* (Python) package[71]. Coverage levels of each gene (in percentage) are calculated from smoothed profiles, for both pulldowns and controls.

For further target identification, transposon-related genes were removed from quantification to avoid bias in the analysis. The fold change between pulldown and control (IP/total) was calculated for each gene profile by calculating the ratio of respective vectors; the fold change of AUC values was also collected for each gene. These fold change gene profiles will further be used in the target identification, and are illustrated in figures (see Figs. 1, 2, and 5; Supplementary Figs. 9, 11, 12) and labeled as "Enrichment (IP/total)". We approached target identification using an FDR calculation based on a true positive set of targets as has been defined for Srp1 (see list in Supplementary Table 1) from the literature. Prior to FDR calculation, the following filtering steps were applied for each gene profile: i) gene coverage is to be set at a minimum of 80%, ii) the AUC-value in the pulldown

condition has to be larger than 5 (i.e. removing signal from lower quantile), and iii) the AUC derived from the fold change profile in the pulldown condition has to at least correspond to the AUC derived from the respective fold change profile in the no-bait wildtype condition (Supplementary Fig. 3). The latter condition puts profiles from pulldown experiments in relation to the no-bait wildtype conditions for the first time in the analysis pipeline. For the FDR calculation, the AUC value per gene and pulldown was scaled relative to the no-bait wildtype condition, i.e. for each gene the AUC-values are summarized for the pulldown and no-bait wildtype experiment; the gene-specific AUC-value in the pulldown is then divided by the latter sum and reflected as a percentage (e.g. shown in Fig. 2a). This no-bait wildtype-scaled AUC-value was then used for calculating a cut-off threshold at FDR 1% (based on the true positive set as mentioned above). At a cutoff threshold of 72%, most of the true positive targets of Srp1 can be recovered (see ROC Curve, Supplementary Fig. 3b). The resulting target set for other pulldowns was then further subjected to manual inspection, and a mean correlation threshold of 0.6 (across replicates) was set for each gene for increasing the quality of hit profiles.

We additionally conducted an analysis using the DESeq2 R-package[72] on the smoothed normalized counts of the genome profiles across pulldowns relative to the no-bait (wildtype) control. For each pulldown analysis, 4 replicates from the selective translatome were considered, as well as the 4 replicates from the no-bait control (factored as conditions). The resulting log2 fold changes and adjusted *P*-values (following the default Benjamini-Hochberg adjustment) are illustrated in Supplementary Fig. 6.

To map SeRP profiles, we extended our previously published scripts[30]. The updated script suit will be made available on Zenodo.

### RIP-qPCR for hit validation

RIP-qPCR was performed as previously described with minor adaptations[30]. Briefly, 400 mL of yeast cultures were set to OD(600) of 0.035 and were grown to an OD(600) of 0.5–0.6 in YPD at 30 °C and 160 rpm. Yeast was harvested by rapid filtration as described above and flash frozen. Cells were lysed in 1.4 mL of high salt lysis buffer (20 mM Hepes-KOH, pH 7.5, 500 mM KCl, 10 mM MgCl₂, 1 mM PMSF, 0.01% IGEPAL, 1 tablet of cOMPLETE protease inhibitor per 50 mL) either containing 0.1 mg/mL CHX or 0.01 mg/mL puromycin (Sigma-Aldrich) using the CryoMill at 30 Hz for 2 min. For the remainder, all buffers either contained CHX or puromycin for the respective samples.

Crosslinking of the sample was conducted as described for SeRP experiments but only using 1.4 mL of buffer to resuspend DSP. Next, lysates were cleared at 15,000 g for 3 min and 20 μL of the cleared supernatant was taken for total RNA extraction. The residual supernatant was applied to 125 μL of equilibrated Streptactin resin supplemented with 60 μL BioLock and 0.1 U/μL Ribolock (Invitrogen). The pulldown, RNA extraction and purification were performed as described above for SeRP.

After the RNA was purified, the concentration of the samples was determined by nanodrop analysis using the absorbance at 260 nm. For subsequent qPCR analysis, 250 ng of RNA was used as input for cDNA synthesis using the VILO Reverse Transcription kit (Invitrogen) according to the manufacturer's procedure. We conducted the additional ezDNase step to remove potential DNA contamination prior to the reverse transcription reaction. qPCR was carried out using the TaqMan Fast and Advanced Master Mix (Applied Biosystems) and purchased and predesigned FAM-MGB-labeled TaqMan probes (Thermo Fisher Scientific; *act1*-mRNA: Sc04120488_s1, *nup60*-mRNA: Sc04098154_s1, *aim44*-mRNA: Sc04171716_s1, *nbp1*-mRNA: Sc04151086_s1, *pop1*-mRNA: Sc04159818_s1). The qPCR was run according to the manufacturer's protocol on a QuantStudio 5 cycler (Applied Biosystems) using the following setting: 50 °C: 2 min, 95 °C: 2 min; 40 cycles: 95 °C: 0:01 min, 60 °C: 0:20 min and quantification at every cycle within the annealing/extension step. The results from each

run were analyzed using QuantStudio analysis software (v1.5.1). Technical triplicates within each biological replicate underwent quality assessment and if needed single technical replicate outliers were omitted. The relative quantification value was determined using the respective input samples (total RNA) for each condition tested.

## Protein analysis

To analyze the bait-StrepII enrichment and the efficiency of crosslinking, samples were mixed with 4 × NuPAGE loading dye containing 5% of beta-mercaptoethanol if not otherwise specified. Samples were boiled at 70 °C for 5 min prior to loading. Loading of the NuPAGE Bis-Tris gels (Invitrogen) was conducted as follows: four microliters of the lysate (~0.08%) for reducing and non-reducing conditions (non-reducing: 4 × NuPAGE loading dye without beta-mercaptoethanol) were loaded to address crosslinking efficiency; four microliters of the resuspended ribosome pellet (0.4%) and 10 μL of the boiled Streptactin resin (equivalent to 4.0%) were loaded to evaluate bait-StrepII enrichment. The samples were run in MOPS buffer (Invitrogen) at 180 V for 50 min.

To analyze the crosslinking efficiency, gels were stained with Instant Blue (abcam) as stated in the manufacturer's protocol.

For the analysis of the bait enrichment, the protein was transferred onto 0.45 μm TransBlot Turbo nitrocellulose (Bio-Rad) using the manufacturer's High MW setting of the TurboBlot (Bio-Rad). Subsequently, membranes were blocked in 5% milk in TBS-T (0.02% Tween-20) for 1 h at room temperature while shaking. Afterward, the membranes were incubated with the monoclonal anti-StrepII antibody (EPR12666; ab180957; Lot. No. GR3212622-7) diluted in 1:5,000 in 5% milk in TBS-T. The membranes were incubated with the primary antibody at 4 °C overnight. Afterward, membranes were vigorously washed three times with TBS-T prior to applying the secondary antibody. The secondary anti-rabbit IgG, HRP-linked antibody (Jackson ImmunoResearch, RRID AB_2313567) diluted 1:10,000 in TBS-T was incubated for 1 h at room temperature while shaking. The previously described washing procedure was repeated and the protein was detected by applying Clarity Max Western ECL solution (Bio-Rad) and imaged using the Chemidoc (Bio-Rad).

## Confocal imaging

Cells were grown overnight in a synthetic complete medium without uracil (SC -Ura; ForMedium) at 30 °C at 180 rpm. Before imaging, cells were diluted to OD(600) 0.2 and again grown under the aforementioned conditions until OD(600) 0.4–0.6. 100 μL of the cell suspension was placed onto Concanavalin A (1 mg/mL, Sigma Aldrich) coated slides, settled for 5 min, and washed three times in SC -Ura.

Live imaging was performed on a laser scanning confocal microscope Leica Stellaris 5. Cells were imaged at room temperature. Images were acquired using a 63×/ 1.2 NA water objective (HC PL APO CS2). Z-stacks were collected using the following image setup: excitation (GFP): 488 nm; emission: 493–591 nm; zoom factor 2, four line averaging, and in bi-directional mode. The microscope was operated using the LAS X software provided by Leica Microsystems CMS GmbH (v.4.4.0.24861). 9–11 images per z-stack were maximum intensity projected using ImageJ (v.1.52). The numbers of cells imaged per NLS-GFP construct and uncropped images can be found in the Source Data file.

## GO enrichment analysis

GO-categories from Cellular Compartment (cc), Molecular Function (mf), and Biological Processes (bp) were extracted from Uniprot (https://www.uniprot.org/), along with mappings of yeast proteins to GO-categories. For the analysis, only GO-terms with more than 100 protein entities were considered. The Fisher enrichment calculation (using the *scipy* Python package) was applied to calculate the odds enrichment of each GO-term within the specific hit set relative to all quantified proteins in the Ribosome Profiling dataset. For visualization, only GO-terms were considered that were significant ($P \leq 0.1$) in at least one pulldown condition. P-values were not adjusted. For visualization, the odds matrices were hierarchically clustered using the Euclidean metric (*seaborn* Python package).

## Analysis of pI-values

Uniprot IDs of the targets and nuclear proteome (GO: 0005634) were fetched on Expasy (https://web.expasy.org/compute_pi/). pI-values were extracted and displayed as a violin plot using GraphPad Prism (v9.0.0). Non-parametric (two-sided) Mann-Whitney *U*-test has been applied to calculate the difference in the pI distributions across conditions. P-values were adjusted using Benjamini-Hochberg.

## Analysis of the domain-onset relationship

Domain architecture of respective proteins was analyzed using Inter-Pro (https://www.ebi.ac.uk/interpro/; release 89.0)[73], and Pfam domain descriptions[74]. Additionally, disorder predictions generated by Inter-Pro were considered. Note that Nup1 and Nup60 were annotated based on their recent domain diagrams presented in Mészáros et al.[75] "Domain boundaries" reflect proteins with 50 amino acids off from the domain end or start. If domains are known for several functions, the domain was assigned to both of its functions. Functions of the respective domains were assigned manually according to the Pfam entry and availability of structures embedded in Pfam.

## Meta-analysis regarding Supplementary Figs. 12, 13

Yeast proteins were mapped to data derived from several publications, including Ghaemmaghami et al., 2003 (protein abundance)[76], Arava et al., 2003 (protein synthesis rates)[77], and Wilmund et al., 2013 (Ssb1/2-dependencies)[48]. For calculations regarding protein synthesis and protein abundance, mapped data was stratified across pulldown conditions. A Kolmogorov-Smirnov test (two-sided) was applied for comparing specific pulldown conditions with the data assigned to all nuclear proteins. P-values were adjusted using Benjamini-Hochberg. For Ssb2 and essentiality mapping, we applied Fisher enrichment calculation- again relative to the nuclear proteome.

## Prediction of NLS motifs using AlphaFold

We searched for putative NLS motifs in the target proteins by running AlphaFold-Multimer[39] structure prediction for Srp1 (AA 70–542, without the autoinhibitory N-terminal peptide) in a complex with consecutive 100 AA. long fragments spanning the target sequences from the N-terminus to residue 550 with 50 AA. overlaps. AlphaFold-Multimer was run with default parameters except the max_recycles parameter set to 12 (to ensure convergence of the modeling) using the AlphaPulldown pipeline[78]. The predictions were scored according to combined ipTM (interface predicted TM-score) and pTM score (predicted TM-score), as returned by AlphaFold, and the top-scoring model for each pair was taken for further analysis (Supplementary Data 1). To calculate the multiple sequence alignment in Fig. 3b, we selected all predictions that (i) exhibited the ipTM+pTM score >0.7, (ii) bound to the canonical NLS-binding site in Srp1, (iii) bound in the N- to C-terminus orientation as known from crystal structures of Srp1-NLS complexes. The selected structures were superposed using UCSF Chimera[79] and the multiple sequence alignment was derived from the superposition. Figures were prepared using UCSF ChimeraX[80] and Jalview[81].

## Metagene profiles

For the generation of metagene profiles, total and IP profiles were normalized, respectively, to the sum of normed counts (as summarized in each profile), correcting for the profile length. All profiles were then scaled to 100% with aggregating values in respective bins by averaging. For visualization purposes, a 95% confidence interval has been applied.

## Multiple sequence alignment

Protein sequences for the 11 importins were retrieved on UniProt. FASTA files were used for multiple sequence alignments Clustal Omega (https://www.ebi.ac.uk/Tools/msa/clustalo/). The dendrogram was generated upon the multiple sequence alignment output in Clustal Omega.

## Quantification and statistical analysis

Significance levels are shown when $*P < 0.05$, $**P < 0.01$, $***P < 0.005$, $****P < 0.001$. For adjustments, we applied the Benjamini-Hochberg method as implemented in the Python *scipy* package. We considered *P*-values as not significant (ns) when $P > 0.05$. All statistical tests were applied in a two-sided manner (if not indicated otherwise) and consider the underlying value distributions. Accordingly, parametric or non-parametric tests have been applied.

Selective ribosome enrichment plots depict data of $n = 4$ replicates for each SeRP experiment. Arbitrary background thresholds indicated as a gray dashed line was set to 1.5 or 2.0, respectively, and are specified in the respective figure legends.

Western Blot analysis to analyze the enrichment of importins and Coomassie gels to check for crosslinking experiments shown in Supplementary Fig. 1c, d were performed for every biologically independent experiment ($n = 4$).

Imaging shown in Fig. 2g, h were repeated at least twice.

## Reporting summary

Further information on research design is available in the Nature Portfolio Reporting Summary linked to this article.

## Data availability

Underlying metadata including uncropped images of Western Blots and Coomassie gels, data used of previously published data (metadata), and data of Ssb1/2[17] and TRiC[16] is available in a Source Data file. The selective ribosome profiling data for all experiments conducted in this study are available on the European Nucleotide Archive database under the accession code PRJEB53855. For the initial processing of the ribosome-protected footprints, we used a script suite for selective ribosome profiling[14]. Furthermore, we provide scripts (https://doi.org/10.5281/zenodo.2602493 and https://doi.org/10.5281/zenodo.7753270), which can be used to process ribosome profiling data and to generate selective ribosome profiles. Note, all selective ribosome profiles for the entire coding genome of *S. cerevisiae* is available at https://doi.org/10.5281/zenodo.7753270. This script suite also provides the required reference genome files for *Saccharomyces cerevisiae* (*S288C_reference_sequence_R64-1-1_20110203.fsa*), as downloaded from SGD [http://sgd-archive.yeastgenome.org/sequence/S288C_reference/genome_releases/S288C_reference_genome_R64-1-1_20110203.tgz]. We also used *sacCer3.ensgene.gtf* for mapping introns and exons, as extracted from Ensembl [https://hgdownload.soe.ucsc.edu/goldenPath/sacCer3/bigZips/genes/sacCer3.ensGene.gtf.gz][69]. Previously published SeRP data for Ssb1/2[17] and Ssb and TRiC[16] can be accessed at Gene Expression Omnibus (GEO) under GEO: GSE93830 and GEO: GSE114882. The structure of the auto-inhibitory NLS of Srp1 bound to Srp1 (Fig. 3a) can be found in the Protein Data Bank (PDB) under the accession number 1WA5 [https://doi.org/10.2210/pdb1WA5/pdb][82]. The ribosome structure (Fig. 4) can be accessed under PDB accession number 4V7R [https://doi.org/10.2210/pdb4V7R/pdb][47]. AlphaFold-Multimer models are available on Zenodo (https://doi.org/10.5281/zenodo.7753270). AlphaFold models of the r-proteins (Fig. 5) originate from the AlphaFold database (https://alphafold.ebi.ac.uk/)[83,84]. Structures can be accessed under accession codes A0A0J9XHQ9 (Rps5), A0A1X7R1F4 (Rpl18a), and A0A0F7RSH3 (Rpl28). Source data are provided with this paper.

## Code availability

The ribosome profiling processing pipeline can be obtained from Zenodo (https://doi.org/10.5281/zenodo.2602493)[14]. Our previously published MatLab scripts are deposited in Zenodo (https://doi.org/10.5281/zenodo.5887402)[30]. The updated version and the SeRP hit identification pipeline is available on Zenodo (https://doi.org/10.5281/zenodo.7753270). AlphaFold modeling of Srp1 bound to NLS was carried out using AlphaPulldown (https://github.com/KosinskiLab/AlphaPulldown)[78].

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

## Acknowledgements
The authors thank Patrick Hoffmann, Filipa Pereira, Katharina Zarnack and Gerhard Hummer for insightful discussions; Stefanie Böhm, Erin Schuman, and Alessandro Ori for critical reading of the manuscript. We would like to thank Dingquan Yu and Jan Kosinski from CSSB & EMBL Hamburg for support with AlphaFold modeling. The authors would like to acknowledge the support of the Imaging Facility at the Max Planck Institute of Brain Research. Additionally, the authors like to thank Georg Stoecklin, Lars Steinmetz, and Britta Brügger for the critical assessment of the project. M.B. acknowledges funding by the Max Planck Society and the European Research Council (724349-ComplexAssembly).

## Author contributions
M.S. conceived the project, designed experiments, performed experiments, analyzed data, and wrote the manuscript. N.R. developed the analysis pipeline, analyzed data, and wrote the manuscript. A.O.-K. conducted AlphaFold modeling, analyzed data, and wrote the manuscript. A.B. performed experiments. J.J.M.L., N.T.D.d.A, and J.P. performed experiments and analyzed data. S.R.N. performed experiments. V.B. designed experiments and supervised the project. M.B. conceived the project, designed experiments, analyzed data, supervised the project, and wrote the manuscript.

## Funding

## Competing interests
The authors declare no competing interests.
