## [Peer Review File · Nature Communications]

REVIEWER COMMENTS

Reviewer #1 (Remarks to the Author):

Seidel et al presents an exciting study of associations of yeast importins with nascent chains during translation. They measured the importin associations of a diverse set of nascent chains using selective ribosome profiling. They found that only a small subset of importins (Srp-Kap95, Kap123 and Kap121) associate with many of their cargoes, and the importin and cargo nascent chain pair remain tethered in a manner resembling profiles for protein complexes. They then looked into associations with Srp-Kap95 and Kap123. They found Srp-Kap95 tethered to the N-terminal portions of many nascent chains that contain classical-NLSs. They also found Kap123 association with the nascent chains of 12 ribosomal proteins at their C-terminal folded domains while nascent chain chaperones Ssb1/2 engaged basic disordered N-terminal regions of the ribosomal protein cargoes. Finally, they found that ribosome occupancy is higher at these lysine- and arginine rich portions of the nascent chains, corresponding with the less abundant lys and arg tRNAs, that in turn give time for importin-binding downstream. This work is exciting, novel and changes the way we think about importin-cargo interactions. I urge publication in Nature Communications upon minor revisions.

Comments:

1) The results and discussion are generally clear and well written but the introduction section needs some editing, both for accuracy and clarity.

a) On line 38, “transportin β 2” should be Transportin-1.

b) The 2nd half of that sentence (line 39) is confusing because it lacks information. Perhaps the authors could replace “and interference with cargo binding causes non-native phase transitions” with “and interference with importin-cargo binding causes cargo self-association and phase transitions”.

c) It is unclear what the authors are trying to say in lines 42-44.

d) On line 47, the use of the word “patches” seems not quite right. Patches is more suitable to describe surfaces. The nascent chain is linear so perhaps basic segments or simply basic residues may be simpler and more appropriate.

e) In lines 52-43, the authors should expand on or clarify the sentence “Our approach led to the identification of a specific subset of importins that co-translationally associate with various cargoes, many of them remained previously unidentified.” The last part of the sentence seems an important point but written as is doesn't do it justice.

f) The next sentence on lines 54-55 is also difficult to understand.

2) The last paragraph of page 10 contains an interesting observation that the C-terminal folded domains of r-proteins that Kap123 interacts with are all buried in the mature ribosome. The last sentence on prevention of ribosome reimport is a good one. But, it is less relevant to translation and nascent chains of r-proteins, which is the focus of the manuscript. Perhaps an additional interpretation is that these Kap123 interaction sites are all macromolecular contact sites that might be prone to inappropriate contacts outside of the context of the ribosome and Kap123-binding prevent the potentially erroneous and toxic interactions.

3) It is unclear what the authors are trying to say with the last sentence of the 1st paragraph of discussion (lines 349-350).

4) In the discussion, the authors raise the interesting point that histones and other r-proteins do not interact co-translationally with importins and that this might be due to the need for the nascent chains to interact with specialized folding chaperone network. It has indeed been worked out for H3 and H4 biogenesis that heat shock proteins and chaperones like NASP1 bind the histones right after or maybe during translation. After all, core histones fold together, H3 with H4 and H2A with H2B into heterodimers, prior to binding their importins. Do reference Campos et al NSMB 2010.

Reviewer #2 (Remarks to the Author):

The manuscript titled "Co-translational binding of importins to nascent proteins" by Maximilian Seidel et al. systematically analyzed co-translational binding of all 11 importins in *Saccharomyces cerevisiae*, utilizing selective ribosome profiling, to capture and characterize importins' nascent-substrate pool. The authors identified a specific subset of importins that co-translationally associate with various cargoes, including many novel cargoes. The authors show that importins act sequentially with the Hsp70 ribosome associated chaperone Ssb1/2 of the , in particular on ribosomal proteins.

The manuscript represents an important contribution to the field of protein biogenesis pathways, suggesting biogenesis of nuclear proteins is intertwined with nascent chain chaperoning to promote the faithful protein localization and prevent aggregation.

We recommend its publication with minor revisions.

The revisions we suggest:

*The authors write that Kap123 and Srp1-Kap95 function as co-translational chaperones. Several importins, among them Imp β –Imp7 heterodimer and Kap β 2, have been shown to function as chaperones in mammalian cell culture and in vitro, preventing aggregation of r-proteins, FUS, etc. (for example: Gou L., et al 2018, Jäkel S., et al., 2002).

The authors show by SeRP that Kap123 and Srp1-Kap95 function in a co-translational manner. However, as Kap123 and Srp1-Kap95 function in prevention of aggregation was not shown, the authors should only discuss their role as putative chaperones.

*DSP cross linker - the use of the cross linker stabilizes various interactions states, including very transient interactions. The authors should discuss the possibility that SeRP experiments without the DSP cross linker or utilizing other cross linkers will report on a different pattern of binding.

*Discuss the classes of importins – the ones that function in a co-translational manner compared the ones that do not, in term of substrate pools, structure, electrostatics etc.

* Trigger factor was shown to wrap/cradle its substrates (Saio T., et al., 2014, Science) the authors should discuss a similar mechanism for co-translational importins.

*Section “Ssb1/2 chaperoning occurs upstream of importin binding”. The authors should specify in the title the relevant importin - Kap123.

*In figure 1b,c the importins should be aligned in the same order to allow a comparison between the Pearson correlation of the area under curve (AUC)-values of the selective ribosome enrichment profiles (IP/total) of 5855 genes quantified across the experiments and the 71 manually curated cargoes.

*Figure 2g,h confocal images of nuclear localization: quantification and statistical significance of nuclear enrichment should be added. Number of cells per experiment should be added to the relevant methods section.

*In figure 4a, the Venn diagram should be modified to reflect no overlap.

*In lines 259-262, the authors note that Rpl28 is different from other r-proteins engaged by Kap123 due to “its unstructured N-terminus”. However, it is not clear what distinguishes Rpl28, given that in lines 250-251, the authors write that “r-proteins typically contain a charged N-terminal patch that appears intrinsically disordered in their primary structure, but is engaged in contacts with rRNA in the mature ribosome”. The authors should discuss this.

*In figure 6d, the local onset taken for comparison should be defined.

Second the statistical significance should be clearly assigned to either Kap123 or Srp1-Kap95. Given that the K and R enrichment presented for Kap123 seems very low. If the enrichment for Kap123 is not significant, or very different in comparison to Srp1-Kap95, the biophysical properties segment (lines 305-307) should be changed accordingly.

Reviewer #3 (Remarks to the Author):

The study by Seidel et al addresses the question whether yeast importins bind their cargo in a cotranslational manner, i.e. while the nascent chain is still tethered at the ribosome. The authors apply selective ribosome profiling, that is an antibody pulldown of the respectively tagged importin followed by a deep sequencing of the ribosome-protected fragments (ribosome footprints). The paper is rich on data (i.e. each sequencing data set is done in four independent biological replicates), however, the data analysis does not convince with the rigor such analysis should be done to substantiate authors hypothesis. Instead of showing global analysis, anecdotal examples are chosen.

Currently, importins are known to posttranslationally bind substrates. Such profoundly new claim, that (all) importins would bind their clients in a translational fashion, should be supported not only by deep sequencing data - needless to mention, in a deep-seq data one can find what you look for – but substantiated with experiments to support the biological meaning (e.g. under depletion conditions to show diminished/decreased cotraslational binding following titration (down or up) of some importins).

Further major points requiring consideration:

1. The IP between different replicates is quite fluctuating, judging by the intensity of the bands in Fig. S1D. This would reflect the quality of the sequencing and consequently the very low Pearson coefficients for some IPs (Fig. S2A). R2 of 0.15-0.3 is quite low. Instead of presenting all data pooled, which is quite inaccurate way to show the quality of the data, the Pearson coefficients (R2 and not R) should be shown for the biological replicates of each importin separately. It may appear, even by this simple rough judgment of the data, that some of the data sets for some importins might be useless.

2. The length of the ribosome footprints representing genuinely translating ribosome is appr 28-29 nt in yeast (PMID: 19213877, PMID: 30686592; PMID: 28193672; PMID: 2850168). Here, however, the total ribosome profiling is depleted of such reads and rather shifts towards longer (35 nt) reads. This is quite strange raising the question as to whether the authors do indeed capture translating ribosomes. To evidence this, they should show the 3nt periodicity in the footprints which the intrinsic reporter of genuinely translating ribosome.

Are the importins crosslinked to the nascent chains actually? Longer reads may mean they are cross-linked to mRNA and hence protecting longer fragments by the ribonuclease digestion? It is crucial to provide an evidence that importins are indeed bound to nascent chains and not to mRNA through mRNA-binding proteins (the latter are their substrates and the longer reads would be indicative that much larger complexes are captured)

3. Along with the anecdotal examples (Fig. 1F), a metaprofile comparison between the IP and total ribosome must be shown.

4. The algorithm used to select the hits (i.e. AUC score) would select too many false-positives. Simply the much higher peaks over initiation and termination would bias the smoothing. Also, it is correct to take only certain equal subsets of read lengths with an inherent 3nt-periodicity in the IP and total ribosome profiling; using the coverage of total reads biases the comparison. For stringent selection, since the approach (crosslinking and pulldown) is inherently prone to capture many false-positives, more than one algorithm should be used for data analysis (see for example PMID: 27487212).

In sum, while the paper presents a quite rich ribosome profiling data, the evidence built only on deep seq data feels too preliminary and fails to convince that importins bind their cargo cotranslationally.

We want to thank all reviewers for the constructive critique and overall positive assessment of our manuscript! Our detailed point by point response follows below. Our responses to the comments are highlighted in blue. Within direct citations of the main text, reference numbers of main text are listed corresponding to the manuscript reference list.

Reviewer #1 (Remarks to the Author):

Seidel et al presents an exciting study of associations of yeast importins with nascent chains during translation. They measured the importin associations of a diverse set of nascent chains using selective ribosome profiling. They found that only a small subset of importins (Srp-Kap95, Kap123 and Kap121) associate with many of their cargoes, and the importin and cargo nascent chain pair remain tethered in a manner resembling profiles for protein complexes. They then looked into associations with Srp-Kap95 and Kap123. They found Srp-Kap95 tethered to the N-terminal portions of many nascent chains that contain classical-NLSs. They also found Kap123 association with the nascent chains of 12 ribosomal proteins at their C-terminal folded domains while nascent chain chaperones Ssb1/2 engaged basic disordered N-terminal regions of the ribosomal protein cargoes. Finally, they found that ribosome occupancy is higher at these lysine- and arginine rich portions of the nascent chains, corresponding with the less abundant lys and arg tRNAs, that in turn give time for importin-binding downstream. This work is exciting, novel and changes the way we think about importin-cargo interactions. I urge publication in Nature Communications upon minor revisions.

1) The results and discussion are generally clear and well written but the introduction section needs some editing, both for accuracy and clarity.

a) On line 38, “transportin β 2” should be Transportin-1.

b) The 2nd half of that sentence (line 39) is confusing because it lacks information. Perhaps the authors could replace “and interference with cargo binding causes non-native phase transitions” with “and interference with importin-cargo binding causes cargo self-association and phase transitions”.

We corrected the respective sentence as follows (line 38-40):

“The importin transportin-1 (TNPO1 or Kap β 2) suppresses phase separation of RNA-binding proteins such as FUS and interference with importin-cargo binding causes cargo self-association and phase transitions.⁸⁻¹¹”

c) It is unclear what the authors are trying to say in lines 42-44.

We edited the sentence as follows: “These previous studies inferred post-translational chaperoning mechanisms for limited number of individual cargoes.”

d) On line 47, the use of the word “patches” seems not quite right. Patches is more suitable to describe surfaces. The nascent chain is linear so perhaps basic segments or simply basic residues may be simpler and more appropriate.

Protein folding can be initiated within the ribosomal exit tunnel such as alpha-helices, hairpins and small helical domains and can further adopt larger structures in co-translational fashion.^{1,2} Hence, the exact three-dimensional structure of a nascent chain may not always be linear. The exact phrasing of ‘basic patches’ was first introduced in Jäkel et al.³ to describe a cluster of

basic residues. We therefore feel that the term of ‘basic segments’ would also not accurately capture the challenge folding potentially sticky basic residues.

e) In lines 52-43, the authors should expand on or clarify the sentence “Our approach led to the identification of a specific subset of importins that co-translationally associate with various cargoes, many of them remained previously unidentified.” The last part of the sentence seems an important point but written as is doesn't do it justice.

Yes, we had phrased this rather cautiously because cargoes have been described by so many individual papers on all kinds of subjects that it is very challenging to examine them comprehensively. To the best of our knowledge of the literature, the SeRP data suggests many previously undescribed Kap123 cargoes, which include Pop1, Pop7, Tgs1, Utp23, Ecm16, Nug1, Ncl1 (ribosome biogenesis factors) and the Nbp1 paralogue Csa1. Among the Srp1-Kap95 cargoes, we first describe the direct targeting of the chromatin remodelers Ino80 and Swr1, Sir4 and additionally ribosome biogenesis factors such as Ssf1, Ssf2, Kri1, Pxr1 and Pus1. In contrast, Kap104 did not reveal any new cargoes but enriched for extremely well studied cargoes such as Nab2 and Syo1.^{4,5}

We have rephrased this sentence as follows: “Our approach led to the identification of a specific subset of importins that co-translationally associate with various cargoes, to the best of our knowledge many of them remained previously unidentified including different ribosome biogenesis factors, cell division machinery and regulators of transcription.”

f) The next sentence on lines 54-55 is also difficult to understand.

We have rephrased the sentence as follows: “We show that nascent chain binding by the chaperone Ssb1/2 frequently precedes cargo recognition by importins, in particular for nucleic acid binding proteins.”

2) The last paragraph of page 10 contains an interesting observation that the C-terminal folded domains of r-proteins that Kap123 interacts with are all buried in the mature ribosome. The last sentence on prevention of ribosome reimport is a good one. But, it is less relevant to translation and nascent chains of r-proteins, which is the focus of the manuscript. Perhaps an additional interpretation is that these Kap123 interaction sites are all macromolecular contact sites that might be prone to inappropriate contacts outside of the context of the ribosome and Kap123-binding prevent the potentially erroneous and toxic interactions.

We have added this as follows: “We propose that this may have two functional benefits. First, Kap123 binding within structured domains may suppress potentially erroneous and toxic interactions of r-proteins outside of their native context. This would be in line with previous reports suggesting that the human homolog of Rps1a/b in complex with polyanions is resolubilized from by the human homolog of Kap123.⁷ Second, the respective sites were engaged in various RNA contacts within the mature 80S ribosome suggesting an inaccessibility for importin binding once ribosome assembly is completed.”

3) It is unclear what the authors are trying to say with the last sentence of the 1st paragraph of discussion (lines 349-350).

We wanted to express that NLSs are kept linear by Ssb1/2 to promote their accessibility to importins. We have edited this sentence as follows:

“This handover might be necessary for cargo recognition by importin in particular for proteins whose importin binding sites become inaccessible in the ternary structure and could be kept unfolded by Ssb1/2 to promote faithful co-translational importin-cargo interaction.”

4) In the discussion, the authors raise the interesting point that histones and other r-proteins do not interact co-translationally with importins and that this might be due to the need for the nascent chains to interact with specialized folding chaperone network. It has indeed been worked out for H3 and H4 biogenesis that heat shock proteins and chaperones like NASP1 bind the histones right after or maybe during translation. After all, core histones fold together, H3 with H4 and H2A with H2B into heterodimers, prior to binding their importins. Do reference Campos et al NSMB 2010.

We have included this as follows: “This may be due to the action of additional and very specialized chaperone networks that may protect them from misfolding in a co-translational fashion.”^{41,61,62}

Reviewer #2 (Remarks to the Author):

The manuscript titled “Co-translational binding of importins to nascent proteins” by Maximilian Seidel et al. systematically analyzed co-translational binding of all 11 importins in *Saccharomyces cerevisiae*, utilizing selective ribosome profiling, to capture and characterize importins’ nascent-substrate pool. The authors identified a specific subset of importins that co-translationally associate with various cargoes, including many novel cargoes. The authors show that importins act sequentially with the Hsp70 ribosome associated chaperone Ssb1/2 of the, in particular on ribosomal proteins.

The manuscript represents an important contribution to the field of protein biogenesis pathways, suggesting biogenesis of nuclear proteins is intertwined with nascent chain chaperoning to promote the faithful protein localization and prevent aggregation.

We recommend its publication with minor revisions.

The revisions we suggest:

*The authors write that Kap123 and Srp1-Kap95 function as co-translational chaperones. Several importins, among them Imp β -Imp7 heterodimer and Kap β 2, have been shown to function as chaperones in mammalian cell culture and in vitro, preventing aggregation of r-proteins, FUS, etc. (for example: Gou L., et al 2018, Jäkel S., et al., 2002).

The authors show by SeRP that Kap123 and Srp1-Kap95 function in a co-translational manner. However, as Kap123 and Srp1-Kap95 function in prevention of aggregation was not shown, the authors should only discuss their role as putative chaperones.

We agree that the chaperoning function of Kap123 and Srp1-Kap95 has not been studied comprehensively, and to the best of our knowledge not at all in yeast. However, we want to draw the attention of the reviewer to the fact that the human homologue of one interaction pair captured by our screen have been analyzed by Jäkel et al. Furthermore, it has been recently shown that that the vertebrate importin α/β complex inhibits phase separation for TPX2.⁶

To make this more transparent, we have rephrased the text as follows: “... Kap123 binding within structured domains may suppress potentially erroneous and toxic interactions of r-proteins outside of their native context. This would be in line with previous reports suggesting

that the human homolog of Rps1a/b in complex with polyanions is resolubilized from by the human homolog of Kap123.⁷”

and

“... Similar chaperoning mechanisms may be relevant for TDP-43, TAF15, EWSR1, hnRNPA1, hnRNPA2, arginine-rich proteins and the spindle assembly factor TPX2.^{10,12,13}”

*DSP cross linker - the use of the cross linker stabilizes various interactions states, including very transient interactions. The authors should discuss the possibility that SeRP experiments without the DSP cross linker or utilizing other cross linkers will report on a different pattern of binding.

Judith Frydman’s group has previously introduced DSP crosslinking to SeRP methods. Stein et al⁷ have done the respective experiments. There is no evidence that crosslinking could potentially alter binding patterns (please refer to **Supplementary Fig. 1K** in <https://ars.els-cdn.com/content/image/1-s2.0-S1097276519304939-mmc4.pdf>). They rather show that DSP crosslinking increases reproducibility of the experiments. We have included the following sentence into the main text to make this more transparent: “Previous systematic analysis has demonstrated that the stabilization of transient interactions by DSP increases reproducibility across replicates but does not affect chaperone binding patterns.¹⁶”

*Discuss the classes of importins – the ones that function in a co-translational manner compared the ones that do not, in term of substrate pools, structure, electrostatics etc.

We have added the following text into the discussion: “It remains yet unclear why only some but not all importins act co-translationally. Although the sequence conservation is considerably low across the 11 yeast importins, they are unified by their low isoelectric point (pI=4.0-5.0), helical HEAT-repeat rich solenoid or superhelical structures and a negatively charged NLS binding pocket.⁵ Interestingly, the importins detected as co-translational binders by our study frequently bind to nuclear proteins that are important for maintaining cellular viability under optimal growth conditions. In contrast, cargoes that were previously described for the other subset of importins, but remained undetected in our experiments, frequently form import complexes with co-chaperones or transcription factors that are activated or translocated into the nucleus upon stress conditions, e.g. Kap114 that is indispensable under saline stress.⁶⁴ It will thus be interesting to investigate such interactions under permissive conditions in the future.”

* Trigger factor was shown to wrap/cradle its substrates (Saio T., et al., 2014, Science) the authors should discuss a similar mechanism for co-translational importins.

We have added the following to the main text: “Furthermore, the structure of some importins, in particular Kap123, are shaped such that they may warp around their cargoes to protect them from their environment.^{54,55} This binding mode could be reminiscent to the activity of trigger factor that co-translationally cages substrates to regulate aggregation-prone regions.⁵⁶”

*Section “Ssb1/2 chaperoning occurs upstream of importin binding”. The authors should specify in the title the relevant importin - Kap123.

We have changed the title as suggested: “Ssb1/2 chaperoning occurs upstream of Kap123 binding.”

*In figure 1b,c the importins should be aligned in the same order to allow a comparison between the Pearson correlation of the area under curve (AUC)-values of the selective ribosome

enrichment profiles (IP/total) of 5855 genes quantified across the experiments and the 71 manually curated cargoes.

We have adjusted the figure accordingly.

*Figure 2g,h confocal images of nuclear localization: quantification and statistical significance of nuclear enrichment should be added. Number of cells per experiment should be added to the relevant methods section.

We appreciate this comment and attempted to do this, but it turned out non-trivial. To quantify the ratio of nuclear to cytoplasmic signal, one would have to robustly segment the nuclei in the negative control cells expressing GFP, but this is not straight forward because the signal simply looks smooth. We however think that the difference to NLS fusion constructs is rather obvious. We want to stress that clear signal occurs in the vast majority of cells except for some that may be out of focus. The images have not been ‘cherry picked.’ For clarity, uncropped images will be made available in the Source Data file. As suggested, we have counted all imaged cells and listed them in the Source Data file alongside the uncropped images. Generally, between 100-200 cells were imaged.

*In figure 4a, the Venn diagram should be modified to reflect no overlap.

We have adjusted the **Figure 4a** as requested.

*In lines 259-262, the authors note that Rpl28 is different from other r-proteins engaged by Kap123 due to “its unstructured N-terminus”. However, it is not clear what distinguishes Rpl28, given that in lines 250-251, the authors write that “r-proteins typically contain a charged N-terminal patch that appears intrinsically disordered in their primary structure, but is engaged in contacts with rRNA in the mature ribosome”. The authors should discuss this.

We apologize for this inaccuracy and want to clarify that the distinguishing feature is that almost no signal is observed for Ssb1/2 binding rephrased as follows: “A notable exception was Rpl28 that may not require Ssb1/2-binding (**Fig. 4d** and **5c**).”

*In figure 6d, the local onset taken for comparison should be defined.

Second the statistical significance should be clearly assigned to either Kap123 or Srp1-Kap95. Given that the K and R enrichment presented for Kap123 seems very low. If the enrichment for Kap123 is not significant, or very different in comparison to Srp1-Kap95, the biophysical properties segment (lines 305-307) should be changed accordingly.

If the enrichment for Kap123 is not significant, or very different in comparison to Srp1-Kap95, the biophysical properties segment (lines 305-307) should be changed accordingly.

For **Fig. 6d**, we have considered amino acid stretches -80 to -30 codons away from the actual onset. For each amino acid, we then calculated an enrichment in those very stretches relative to the entire protein sequence, for Kap123 and Srp1-Kap95 hits, respectively. Hence, the *p*-values are specific for each amino acid and each target set; the *p*-values do not relate to any comparison between Kap123 and Srp1-Kap95 enrichment values. The figure legend indicates all specific *p*-values and always indicates the respective importin. With regard to the K and R enrichment, we indeed observe a significant enrichment of these amino acids in Srp1/Kap95-targets prior to onset, but this is not the case for Kap123. However, Kap123 targets are generally more K/R rich than control nuclear proteome or other co-translationally bound cargo by other importins. We rephrased the segment for more clarity: “The local amino acid signature upstream of the

observed onsets for Srp1 displays a significant enrichment for the positively charged residues lysine and arginine. This enrichment appears less accentuated for Kap123 (**Fig. 6d**). However, this may be due to the generally high lysine and arginine content within its cargoes (**Fig. S13d**).”

Reviewer #3 (Remarks to the Author):

The study by Seidel et al addresses the question whether yeast importins bind their cargo in a cotranslational manner, i.e. while the nascent chain is still tethered at the ribosome. The authors apply selective ribosome profiling, that is an antibody pulldown of the respectively tagged importin followed by a deep sequencing of the ribosome-protected fragments (ribosome footprints). The paper is rich on data (i.e. each sequencing data set is done in four independent biological replicates), however, the data analysis does not convince with the rigor such analysis should be done to substantiate authors hypothesis. Instead of showing global analysis, anecdotal examples are chosen.

Currently, importins are known to posttranslationally bind substrates. Such profoundly new claim, that (all) importins would bind their clients in a translational fashion, should be supported not only by deep sequencing data - needless to mention, in a deep-seq data one can find what you look for – but substantiated with experiments to support the biological meaning (e.g. under depletion conditions to show diminished/decreased cotraslational binding following titration (down or up) of some importins).

In sum, while the paper presents a quite rich ribosome profiling data, the evidence built only on deep seq data feels too preliminary and fails to convince that importins bind their cargo cotranslationally.

We want to thank the reviewer for his/her critical assessment and for acknowledging that our study is ‘rich in data’. We highly appreciated the reviewer’s suggestions to make the data analysis more understandable and transparent and have included a number of additional analyses into the manuscript. We respectfully disagree with a few individual statements made by the reviewer, as explained in detail below.

We want to clarify that our experiments are not based on “antibody pulldown” as stated by the reviewer. Instead, we rather rely on endogenous protein purification of StrepII-tagged proteins. This approach is advantageous because of its high affinity and specificity and allows to enrich even for small quantities of importins from ribosome nascent chain complexes on a much more rapid time scale.

We want to clarify that we did not claim “that (all) importins would bind their clients in a translational fashion”. On the contrary, although our screen targets all importins systematically, not all of them were found to co-translationally bind to cargo, but only a very specific subset. This subset happens to be the same that has been previously identified to act as chaperones for nucleic acid binders^{3,9–11}, although post-translationally.

We want to stress that the technical standard of our study goes clearly beyond the established state of the art in the field. While previous studies by prominent laboratories^{7,12–18} relied on two biological replicates, we consistently used four. To the best of our knowledge, no previous study has estimated a false positive discovery rate based on a known ground truth. Here, we take advantage of the fact that substrate spectrum of the Srp1/Kap95 dependent import pathway has been well characterized by different methods. Our study further harnesses the power of the many, complimentary experiments across all importins, which suggest that identified cargoes are unique within individual experiments.

We strongly disagree with the very general statement of the reviewer that “in a deep-seq data one can find what you look for”. We want to stress that our study is well controlled. The hits

show a strong signal that is significantly elevated over controls (non-tagged wildtype strains) across multiple replicates. Not only do they recover known nuclear import substrates as pointed out above, the observed position of the binding onsets within the ORFs are consistent with known domain structures, e.g. the IBB domain of Srp1 that binds to Kap95 (**Fig. 2c** highlighting the recognition of the autoinhibitory NLS of nascent Srp1 by Kap95, and **Supplementary Table 2**). We further want to highlight that we provide substantial evidence that SeRP determined interaction sites are in line to annotated nuclear localization sequences (NLS) from literature, are capable to reveal novel NLSs (**Fig. 2g, 2h**), and yield high confidence scores in alpha fold models of the respective interactions, that are consistent with the known NLS binding pocket and bipartite arrangement (**Fig. 3**).

In response to the statement that “Instead of showing global analysis, anecdotal examples are chosen” we wish to point out that global analysis is shown in **Fig. 1b, 1c, 2a, 3, 6** and at least 6 supplementary display items. The identified cargos and their corresponding onsets are globally assessed in particular to derive general patterns within our data. We have highlighted specific examples in the manuscript, (e.g. **Fig. 1f**) to illustrate the functionality of our analysis pipeline and to highlight the precision of our data. We do not think that this is an unusual approach. We nevertheless followed a number of specific suggestions made by the reviewer that were indeed very useful and included additional analysis (see below).

The added benefit of the titration experiments suggested by the reviewer is not clear to us. Yes, a change in concentration of the bait protein is expected to result in a change of signal intensity. But we have included non-tagged wildtype strains as control and already shown that this abolishes the signal.

Further major points requiring consideration:

1. The IP between different replicates is quite fluctuating, judging by the intensity of the bands in **Fig. S1D**. This would reflect the quality of the sequencing and consequently the very low Pearson coefficients for some IPs (**Fig. S2A**). R2 of 0.15-0.3 is quite low. Instead of presenting all data pooled, which is quite inaccurate way to show the quality of the data, the Pearson coefficients (R2 and not R) should be shown for the biological replicates of each importin separately. It may appear, even by this simple rough judgment of the data, that some of the data sets for some importins might be useless.

The Western Blots shown in **Fig. S1d**, were generated from boiled Streptactin beads of our affinity purifications. We included them to demonstrate that we consistently enrich the bait protein as compared to the input ribosome fraction. However, those samples were not normalized by BCA, and this is also not necessary to demonstrate successful affinity purification. Hence some variations in the protein detection are expected.

With regard to the Pearson correlation coefficients, we have now applied the Pearson correlation coefficient on all smoothed gene profiles across all replicates to account for any sparsity issues as we might have encountered them in the previous version of the manuscript when correlating experiments. We have followed the reviewer’s recommendation to calculate and demonstrate these values for the biological replicates of each importin separately. This is now visible in **Fig. S2a**, which specifically focuses on the IP experiments. As is also summarized in **Fig. S2c**. The median correlation of the IP biological replicates is higher than 0.9. Even across biological replicates, the median correlation value is above 0.7. We also highlight in **Fig. S2b**, how total replicates cluster relative to the IP replicates in each biological experiment. It becomes clear that in case of Kap123 – given that this experiment produced most

of significant onset hits – total and IP differ more drastically than in wildtype or other experiments where not as many hits were identified. We want to remind the reviewer that only a subset of importins contains a larger number of hits, while the data for the other importins that do not show any co-translational association is not expected to be different from controls samples.

2. The length of the ribosome footprints representing genuinely translating ribosome is appr 28-29 nt in yeast (PMID: 19213877, PMID: 30686592; PMID: 28193672; PMID: 2850168). Here, however, the total ribosome profiling is depleted of such reads and rather shifts towards longer (35 nt) reads. This is quite strange raising the question as to whether the authors do indeed capture translating ribosomes. To evidence this, they should show the 3nt periodicity in the footprints which the intrinsic reporter of genuinely translating ribosome.

Inspired by previous publications from prominent laboratories,^{7,16,19} we have used the well-established crosslinker DSP in our experiments. It was not only shown to be compatible to ribosome profiling experiments and but also to increase reproducibility. Please refer to **Supplementary Fig. 1K** in Stein et al.⁷ (<https://ars.els-cdn.com/content/image/1-s2.0-S1097276519304939-mmc4.pdf>).

However, crosslinking may alter ribosome footprint distributions by stabilizing the interface between ribosomal subunits.²⁰

We agree that ribosomal footprints reported without cross-linker usually show a median length of 30 nt. Ours are shifted towards a median length of 33 nt (not 35 nt). However, footprints can be representatives of variable structural states of the ribosomes. Inhibitors such as anisomycin (ANS) yield ribosome footprints from ribosomes with an open A site, which shifts the media footprint length to 21 nt.²¹ Our cross-linker likely alters the accessibility of ribosome protected mRNA for RNase I.²⁰ Due to the variability of the structural states of cross-linked ribosomes and read length, the individual reads cannot consistently register with the A site. As a consequence, it is not even expected to observe the 3 nt periodicity in this data set.

However, as exemplified below, we do observe a strong enrichment of ribosomal footprints within the open reading frame (ORF) contrasting the neighboring UTRs. An enrichment at the start codon is observed. This is clear cut evidence that our data represents direct importin-nascent chain interactions at translating ribosomes. Moreover, to confirm that importins enrich footprints for their cargo through a co-translational interaction, we conducted RIP-qPCR^{15,22–24} for Kap123 and some of its cargoes and found a translation-dependent decrease of the enrichment upon puromycin treatment (**Fig. S8**).

Figure 1: 3 nt periodicity plot for Kap123 total translatoome footprints. Position 0 corresponds to the start codon. Region from -10 to 0 corresponds to 5'UTR. Periodicity of the ORF is shown upstream of 0.

3. Are the importins crosslinked to the nascent chains actually? Longer reads may mean they are cross-linked to mRNA and hence protecting longer fragments by the ribonuclease digestion? It is crucial to provide an evidence that importins are indeed bound to nascent chains and not to mRNA through mRNA-binding proteins (the latter are their substrates and the longer reads would be indicative that much larger complexes are captured).

In addition to the arguments and additional experiments pointed out in response to the reviewers point 2, we want to stress the following:

The use of DSP was thoroughly addressed for its application in selective ribosome profiling^{7,19} including comparative experiments with and without cross-linker. It was not found to induce false positive hits.

DSP is a primary amine specific crosslinker. While, adenine, guanosine and cytosine contain amine groups, these groups do not act as primary amines due to the tautomerism within the nucleobases. Therefore, crosslinking of RNA to protein is chemically unfavorable. Harsher crosslinking conditions are required to stably capture protein-RNA interactions such as formaldehyde or UV treatment.

The subgroup of importins that are chaperones are highly acidic (pI= 4.0-5.0)²⁵ and mimic the properties of RNA. The negative charges promote the binding of importins for positively charged binding sites of RBPs to suppress RNA binding in a mutually exclusive manner. Based on these electrostatic and biochemical properties, it is very difficult to conceive that importins are cross-linked to RNA.

Moreover, ribosome-nascent chain complexes are purified after the RNase I treatment by a 20 % sucrose cushion. The putative importin•RBP•RNA complexes the reviewer speculates about should shift within the cushion towards the surface as they become disconnected from heavier ribosomes. Novel techniques such as Grad-Seq highlight, that ribosome-associated binders such as the co-translational chaperone GroEL are not effected by RNase I digest suggesting direct ribosome tethering.^{26,27}

At last, the observed onset curves are consistent with biochemical and structural data thus arguing that we capture importin binding to nascent chains. Specifically, some of the identified hits for Srp1-Kap95, Kap123, Kap121 and Kap104 correspond to previously described cargo of these importins and their corresponding NLS (**Table S2** and Chook and Suel (2010)²⁵). This

should not be observable if importins were cross-linked to RNA. Further, GFP-fusions of such NLSs have led to nuclear localization further strengthening that the data corresponds to protein-protein interactions of importins to their cargo. We have confirmed some of the hits of Kap123 using RIP-qPCR^{15,22-24} in combination with translation specific drugs and show that indeed interactions are established co-translationally by an interaction of an importin with the nascent chain.

3. Along with the anecdotal examples (Fig. 1F), a metaprofile comparison between the IP and total ribosome must be shown.

We included metagene profiles in for all experiments **Fig. S4a** and curated hits (**Fig. S4b**).

4. The algorithm used to select the hits (i.e. AUC score) would select too many false-positives. Simply the much higher peaks over initiation and termination would bias the smoothing. Also, it is correct to take only certain equal subsets of read lengths with an inherent 3nt-periodicity in the IP and total ribosome profiling; using the coverage of total reads biases the comparison. For stringent selection, since the approach (crosslinking and pulldown) is inherently prone to capture many false-positives, more than one algorithm should be used for data analysis (see for example PMID: 27487212).

We agree with the reviewer that if the AUC score would be the sole metric used for selecting hits, we would indeed capture too many false positives. However, we want to highlight to the reviewer that we have considered that in our pipeline (**Fig. S3**) by including a number of filters, including fold-change thresholds for IP vs. total and bait vs. not bait. But what is perhaps the most important part of the pipeline is the fact that the FDR-calculation is not applied on the AUC-value in the pulldown-experiment directly, but relative to the wildtype experiment (denoted as “Relative AUC (pulldown vs. WT)” in **Fig. S3**). This is important, because – as the reviewer correctly points out – there are indeed higher peaks over initiation and termination, but these signals can be removed as noise that is occurring in control experiments as well.

Given the nature of (expected) importin binding (onset with persistently elevated profile), we tuned our pipeline to dismiss little local effects (e.g. at 3nt-granularity) to avoid picking up hits with “spikes” in their profiles. Hence, we applied a stringent filter on coverage as well.

Finally, we only present potential targets a hits in the manuscript once they have passed a rigorous manual control as well.

We hope that the above arguments convince the reviewer that the approach presented, i.e. the combination of an FDR-calculation on relative AUC (pulldown vs. wildtype) and a manual check, is rigorous in picking up target hits and in fact reduces the number of false positives. In fact, this approach enables benchmarking according to the rate of false positives allowed (e.g. 1-5% FDR).

As an orthogonal approach, we show the results of our DESeq2-analysis for the pulldowns that provided most of the clear hits (Srp1, Kap95 and Kap123) in **Fig. S6**. The colored points signify nucleotide positions of cargo genes (as were identified by the above pipeline), indicating that DESeq2 could indeed recover the cargos as we have identified them in our AUC-approach as well. However, as DESeq2 is applied at the nucleotide level, it tends to also pick up hits that are characterized by local “spikes”. For this reason, we had decided to develop a pipeline that is picking up profiles in a manner to avoid that, and can be tuned to an FDR level.

References

1. Rodnina, M. V. & Wintermeyer, W. Protein Elongation, Co-translational Folding and Targeting. *J. Mol. Biol.* **428**, 2165–2185 (2016).
2. Kramer, G., Shiber, A. & Bukau, B. Mechanisms of cotranslational maturation of newly synthesized proteins. *Annu. Rev. Biochem.* **88**, 337–364 (2019).
3. Jakel, S., Mingot, J.-M., Scharzmaier, P., Hartmann, E. & Görlich, D. Importins fulfil a dual function as nuclear import receptors and cytoplasmic chaperones for exposed basic domains. *EMBO J.* **21**, 377–386 (2002).
4. Lee, D. C. Y. & Aitchison, J. D. Kap104p-mediated nuclear import. *J. Biol. Chem.* **274**, 29031–29037 (1999).
5. Pausch, P. *et al.* Co-translational capturing of nascent ribosomal proteins by their dedicated chaperones. *Nat. Commun.* **6**, 1–15 (2015).
6. Safari, M. S., King, M. R., Brangwynne, C. P. & Petry, S. Interaction of spindle assembly factor TPX2 with importins- α/β inhibits protein phase separation. *J. Biol. Chem.* **297**, 100998 (2021).
7. Stein, K. C., Kriel, A. & Frydman, J. Nascent Polypeptide Domain Topology and Elongation Rate Direct the Cotranslational Hierarchy of Hsp70 and TRiC/CCT. *Mol. Cell* **75**, 1117–1130.e5 (2019).
8. Wing, C. E., Fung, H. Y. J. & Chook, Y. M. Karyopherin-mediated nucleocytoplasmic transport. *Nat. Rev. Mol. Cell Biol.* **0123456789**, (2022).
9. Guo, L. *et al.* Nuclear-Import Receptors Reverse Aberrant Phase Transitions of RNA-Binding Proteins with Prion-like Domains. *Cell* **173**, 677–692.e20 (2018).
10. Hofweber, M. *et al.* Phase Separation of FUS Is Suppressed by Its Nuclear Import Receptor and Arginine Methylation. *Cell* **173**, 706–719.e13 (2018).
11. Yoshizawa, T. *et al.* Nuclear Import Receptor Inhibits Phase Separation of FUS through Binding to Multiple Sites. *Cell* **173**, 693–705.e22 (2018).
12. Döring, K. *et al.* Profiling Ssb-Nascent Chain Interactions Reveals Principles of Hsp70-Assisted Folding. *Cell* **170**, 298–311.e20 (2017).
13. Chartron, J. W., Hunt, K. C. L. & Frydman, J. Cotranslational signal-independent SRP preloading during membrane targeting. *Nature* **536**, 224–228 (2016).
14. Schibich, D. *et al.* Global profiling of SRP interaction with nascent polypeptides. *Nature* **536**, 219–223 (2016).
15. Shiber, A. *et al.* Cotranslational assembly of protein complexes in eukaryotes revealed by ribosome profiling. *Nature* **561**, 268–272 (2018).
16. Oh, E. *et al.* Selective ribosome profiling reveals the cotranslational chaperone action of trigger factor in vivo. *Cell* **147**, 1295–1308 (2011).
17. Eismann, L. *et al.* Selective ribosome profiling reveals a role for SecB in the cotranslational inner membrane protein biogenesis. *Cell Rep.* **41**, 111776 (2022).
18. Wu, C. C. C., Peterson, A., Zinshteyn, B., Regot, S. & Green, R. Ribosome Collisions Trigger General Stress Responses to Regulate Cell Fate. *Cell* **182**, 404–416.e14 (2020).
19. Becker, A. H., Oh, E., Weissman, J. S., Kramer, G. & Bukau, B. Selective ribosome profiling as a tool for studying the interaction of chaperones and targeting factors with nascent polypeptide chains and ribosomes. *Nat. Protoc.* **8**, 2212–2239 (2013).
20. Tüting, C., Iacobucci, C., Ihling, C. H., Kastritis, P. L. & Sinz, A. Structural analysis of 70S ribosomes by cross-linking/mass spectrometry reveals conformational plasticity. *Sci. Rep.* **10**, 1–13 (2020).
21. Wu, C. C. C., Zinshteyn, B., Wehner, K. A. & Green, R. High-Resolution Ribosome Profiling Defines Discrete Ribosome Elongation States and Translational Regulation during Cellular Stress. *Mol. Cell* **73**, 959–970.e5 (2019).
22. Duncan, C. D. S. & Mata, J. Widespread cotranslational formation of protein complexes.

- PLoS Genet.* **7**, (2011).
23. Seidel, M. *et al.* Co-translational assembly orchestrates competing biogenesis pathways. *Nat. Commun.* 1–15 (2022) doi:10.1038/s41467-022-28878-5.
 24. Kamenova, I. *et al.* Co-translational assembly of mammalian nuclear multisubunit complexes. *Nat. Commun.* **10**, 25–28 (2019).
 25. Chook, Y. M. & Süel, K. E. Nuclear import by karyopherin- β s: Recognition and inhibition. *Biochim. Biophys. Acta - Mol. Cell Res.* **1813**, 1593–1606 (2011).
 26. Gerovac, M. *et al.* Global discovery of bacterial RNA-binding proteins by RNase-sensitive gradient profiles reports a new FinO domain protein. *Rna* **26**, 1448–1463 (2020).
 27. Gerovac, M., Vogel, J. & Smirnov, A. The World of Stable Ribonucleoproteins and Its Mapping With Grad-Seq and Related Approaches. *Front. Mol. Biosci.* **8**, (2021).

REVIEWERS' COMMENTS

Reviewer #3 (Remarks to the Author):

The authors have revised the manuscript and addressed the comments from the first reviewing round.

The 3nt periodicity, along footprint distribution, is an important quality control and must be included in the supplementary file (e.g. Fig. S1). Thereby, in the text should fairly be stated that the ribosome profiling data lacks the commonly observed 3nt-periodicity. It is important to show the whole plot including the 3'UTRs to show the fall off of the footprints and the ability to capture the open-reading frame.

Rebuttal Letter

We would like to thank reviewer 3 for the helpful critical suggestion and have implemented this into our final version of the manuscript.

Reviewer #3 (Remarks to the Author):

The authors have revised the manuscript and addressed the comments from the first reviewing round. The 3nt periodicity, along footprint distribution, is an important quality control and must be included in the supplementary file (e.g. Fig. S1). Thereby, in the text should fairly be stated that the ribosome profiling data lacks the commonly observed 3nt-periodicity. It is important to show the whole plot including the 3'UTRs to show the fall off of the footprints and the ability to capture the open-reading frame.

We have included the 3 nt periodicity plot in the **Supplementary Fig. 1f** for the 5' and 3' end of the ORF. We refer to it in the main text as follows:

“As expected, the footprints are much more prominent within the ORF in comparison to the respective 5' and 3' UTRs, showing that we have captured footprints from translating ribosomes on the respective mRNAs (**Supplementary Fig. 1f**). We note that the 3 nucleotide (nt) periodicity was blurred as compared to previous ribosome profiling experiments that did not use cross-linking.^{1,2} This may be explained by a reduced accessibility of the mRNA for RNase I due to sterically hindrance by the cross-linker, which is consistent with the slightly increased ribosomal footprint length (Supplementary Fig. 1e), thus preventing an accurate registration of the A-site.”